# Increasing heart vascularisation after myocardial infarction using brain natriuretic peptide stimulation of endothelial and WT1+ epicardial cells

Na Li, Stephanie Rignault-Clerc, Christelle Bielmann, Anne-Charlotte Bon-Mathier, Tamara Déglise, Alexia Carboni[†], Mégane Ducrest, Nathalie Rosenblatt-Velin*

Division of Angiology, Heart and Vessel Department, Centre Hospitalier Universitaire Vaudois and University of Lausanne, Lausanne, Switzerland

**Abstract** Brain natriuretic peptide (BNP) treatment increases heart function and decreases heart dilation after myocardial infarction (MI). Here, we investigated whether part of the cardioprotective effect of BNP in infarcted hearts related to improved neovascularisation. Infarcted mice were treated with saline or BNP for 10 days. BNP treatment increased vascularisation and the number of endothelial cells in all areas of infarcted hearts. Endothelial cell lineage tracing showed that BNP directly stimulated the proliferation of resident endothelial cells via NPR-A binding and p38 MAP kinase activation. BNP also stimulated the proliferation of WT1+ epicardium-derived cells but only in the hypoxic area of infarcted hearts. Our results demonstrated that these immature cells have a natural capacity to differentiate into endothelial cells in infarcted hearts. BNP treatment increased their proliferation but not their differentiation capacity. We identified new roles for BNP that hold potential for new therapeutic strategies to improve recovery and clinical outcome after MI.

*For correspondence:
nathalie.rosenblatt@chuv.ch

Present address: [†]Ecole Polytechnique fédéral de Lausanne, Laussane, Switzerland

Competing interests: The authors declare that no competing interests exist.

## Introduction

Increased vascularisation supports heart recovery after ischemia. The formation of new vessels in the hypoxic area restores blood flow, provides oxygen and nutriments to the surviving cells, and promotes the migration and engraftment of new cells. While angiogenic inhibition contributes to the development of heart failure in cardiac injury animal models, early heart reperfusion or increased angiogenesis improves cardiac function and delays the onset of heart failure in patients suffering from cardiac ischemia (*Shiojima et al., 2005*; *Friehs et al., 2006*; *Tirziu et al., 2007*).

Angiogenesis is the main mechanism of neovascularisation in adult infarcted hearts (*Manavski et al., 2018*; *Li et al., 2019*; *Ray et al., 2010*). The origin of new endothelial cells (i.e. resident or infiltrating) as well as the underlying mechanism leading to their proliferation (partial endothelial-to-mesenchymal transition [EndMT] or not) have long been debated. The current consensus is that after myocardial infarction (MI), angiogenesis in the infarct border zone of the heart occurs by clonal expansion of pre-existing resident endothelial cells with no EndMT mechanism (*Manavski et al., 2018*; *Li et al., 2019*; *He et al., 2017*). However, the molecular pathway involved in myocardial neovascularisation after ischemia remains unknown. Indeed, *Payne et al., 2019* recently demonstrated that the developmental VEGFA-MEF2 pathway, which was thought to be involved, is in fact impaired in adult ischaemic hearts.

Stimulating angiogenesis after MI can improve heart recovery. One complementary therapy could be 're-activating' vasculogenesis (i.e. the differentiation of precursor cells into mature endothelial cells), a mechanism that occurs in the heart during development but is quiescent in adult hearts. Epicardial cells, and more precisely, cells expressing the Wilms' tumour 1 transcription factor (WT1)

migrate from the epicardium to the myocardium during heart development and then differentiate into coronary endothelial cells, pericytes, smooth muscle cells, and even cardiomyocytes after epithelial-to-mesenchymal transition (*Zhou et al., 2008*; *Cano et al., 2016*; *Smits et al., 2018*). Consequently, numerous WT1$^+$ cells are found in foetal and neonatal hearts, whereas only a few cells express WT1 in adult hearts (*Duim et al., 2016*; *Duim et al., 2015*).

In hypoxic adult hearts, numerous proliferating WT1$^+$ cells can be localised in the epicardium near the infarct and border zone, suggesting that hypoxia stimulates either the proliferation of the remaining WT1$^+$ cells or the re-expression of WT1 in cardiac cells (*Duim et al., 2015*; *Balbi et al., 2019*; *Zhou et al., 2012*). WT1$^+$ epicardium-derived cells (EPDCs) remain in a thickened layer on the heart surface without migrating into the myocardium or differentiating into other cell types such as cardiomyocytes or endothelial cells (*Zhou et al., 2011*). Different treatments (e.g. injections of thymosin beta four or human amniotic fluid stem cell secretome) fail to induce WT1$^+$ cell differentiation into endothelial cells in adult hearts after MI (*Balbi et al., 2019*; *Zhou et al., 2012*). Despite enhanced WT1$^+$ cell proliferation and higher vessel density in these treated infarcted hearts, no WT1$^+$ cell differentiation into endothelial cells has been detected, implying that WT1$^+$ EPDCs improve neovascularisation in infarcted hearts via paracrine stimulation.

It is therefore important to identify soluble factors to increase neovascularisation in the heart after MI. For several years, we have studied the role of brain natriuretic peptide (BNP) in the heart during ageing and after ischaemic damage (*Bielmann et al., 2015*). BNP is a cardiac hormone belonging to the natriuretic peptide family along with atrial natriuretic peptide (ANP) and C-type natriuretic peptide (CNP). Although low levels of BNP are co-stored with ANP in atria, high levels of BNP are detected in the ventricle (*Potter et al., 2009*). BNP is mainly secreted in the ventricles by cardiomyocytes, fibroblasts, and endothelial and precursor cells (*Bielmann et al., 2015*; *Potter et al., 2009*; *Rosenblatt-Velin et al., 2016*). BNP binds to guanylyl cyclase receptors, NPR-A and NPR-B, thus increasing the intracellular cGMP level (*Potter et al., 2009*).

BNP is synthesised in the cell cytoplasm as pre-proBNP precursor, cleaved by furin and corin into proBNP peptide, and then into the biologically active carboxy-terminal BNP peptide (active BNP) and the inactive N-terminal fragment (NT-proBNP) (*Yandle and Richards, 2015*; *Clerico et al., 2012*). ProBNP, active BNP, and NT-proBNP peptides are continuously secreted by cardiac cells and ProBNP peptide is the major circulating form of BNP in the plasma of healthy individuals (*Shimizu et al., 2002*). O-glycosylated proBNP was also detected in the plasma of patients suffering from heart failure (*Volpe et al., 2016*), and O-glycosylation of proBNP prevents cleavage of this peptide (*Volpe et al., 2016*). Thus, the balance between proBNP and active BNP is impaired in patients with heart diseases, as they have higher levels of inactive proBNP and reduced levels of active BNP. Patients with heart disease therefore have a deficit in functional active BNP (*Chen, 2007*).

BNP secreted by cardiac cells acts on different organs such as the kidneys (modulating sodium and water excretion), vessels (increasing dilation), fat (increasing lipolysis), and pancreas (modulating insulin secretion) (*Rosenblatt-Velin et al., 2016*; *Volpe et al., 2016*). In the heart, a majority of cardiac cells (cardiomyocytes, fibroblasts, endothelial cells) express BNP receptors in physiological and pathological states (*Bielmann et al., 2015*). BNP treatment after injury protects the heart by reducing fibrosis, cardiomyocyte death, and hypertrophy (*D'Souza and Baxter, 2003*; *Moilanen et al., 2011*; *Gorbe et al., 2010*; *Scott et al., 2009*; *Sun et al., 2010*; *Wu et al., 2009*).

In the last few years, we and others reported that BNP supplementation after ischaemic damage promotes the recovery of cardiac function and prevents cardiac remodelling in adult rodent ischaemic hearts (*Bielmann et al., 2015*; *Moilanen et al., 2011*). Furthermore, in clinic, a treatment (LCZ696 or Entresto, Novartis) based on inhibition of neprilysin, an enzyme involved in the degradation of the natriuretic peptides, leads to reduced rate of mortality in patients suffering from heart failure with reduced and preserved cardiac contractility or ejection fraction (*McMurray et al., 2014*). The cellular mechanisms by which BNP exerts its cardioprotective effect are not fully elucidated (*Bielmann et al., 2015*; *Rosenblatt-Velin et al., 2016*; *Moilanen et al., 2011*). Curiously, we observed that BNP stimulates in vitro and in vivo (i.e. in adult infarcted hearts) the proliferation of cardiac non-myocyte cells (NMCs) expressing the stem cell antigen-1 (Sca-1) (*Rignault-Clerc et al., 2017*). As Sca-1$^+$ cells in adult hearts were reported to be pure endothelial cells (*Zhang et al., 2018*), we questioned whether BNP modulates endothelial cell fate.

Thus, since high BNP levels in plasma are associated with increased collateral development in patients with coronary artery disease (*Xi et al., 2011*), this work aimed to determine whether part of the cardioprotective effect of BNP treatment after experimental MI involves increased neovascularisation.

## Results

### BNP direct action on cardiac NMCs after intraperitoneal injections

MI was induced in mice by permanent ligation of the left anterior descending artery. Injection of BNP was immediately performed after surgery and then every 2 days for up to 10 days after surgery (*Figure 1A*). BNP injections may have systemic effects and affect several cell types expressing its receptors (such as fibroblasts, immune cells, endothelial cells in all organs and even cardiomyocytes). In previous work, we first controlled that with the doses of BNP used, we had no effect on systolic blood pressure (measured every day) (*Bielmann et al., 2015*). Secondly, we evaluated cardiac parameters and heart structure by echocardiography in BNP and saline-treated 'unmanipulated' and infarcted mice. No difference was detected in heart rate, cardiac output, left ventricular diastolic volume (LV Vol;d) as well as for the index of systemic vascular resistance between both groups, demonstrating that BNP treatment (at the dose used in this study) had no significant effect on volumia and no peripheral vascular effect (*Bielmann et al., 2015*). Third, contractility (EF@LV Vol;d) was twofold increased 4 weeks after MI in BNP-treated groups and BNP injections reduced clearly heart remodelling (which is the percentage of changes of the left ventricle volume) 1 (−45%, p=0.06) and 4 weeks (−79%, p=0.04) after MI (*Bielmann et al., 2015*). Accordingly, BNP treatment of infarcted hearts reduced also mRNA levels coding for vimentin in the RZ of infarcted hearts 1 (−59%, p=0.004) and 4 (-36%, p=0.025) weeks after MI, which suggests an effect on fibrosis development as already demonstrated by other (*Moilanen et al., 2011*). Finally, BNP treatment reduced also the infarct size (determined by echocardiography analysis with the % of MI length LA) by 15 and 20% 1 and 4 weeks after MI, respectively (results however not statistically different).

Altogether, these results clearly demonstrated that most BNP effects after MI depend on cardiac rather than on vascular effects.

As BNP modulates the function of different organs (kidneys, vessels, pancreas), its effect on hearts could be indirect. Thus, we first aimed to determine whether intraperitoneally injected BNP acted directly or indirectly on cardiac cells.

For this purpose, activations of different components of the BNP signalling pathway were evaluated after BNP injections in unmanipulated or infarcted mice (*Figure 1*). BNP binding to the receptors NPR-A and NPR-B, but not NPR-C, increases intracellular cyclic guanosine 3′,5′-monophosphate (cGMP) levels. cGMP modulates the protein kinase G activity and induces phosphorylation at Ser16 of phospholamban (PLB) (*Rosenblatt-Velin et al., 2016*). Increased intracellular cGMP levels simultaneously activate GMP-related cation channels, leading to cGMP extrusion from the cytoplasm into the circulation.

BNP acts directly on the heart. Indeed, very rapidly after intraperitoneal (ip) BNP injections (i.e. 30 min), increased cGMP levels were detected by ELISA in cardiac tissue (x 3.13, p=0.006) (*Figure 1B*). In the plasma of unmanipulated mice, cGMP concentration increased 3.8-fold (103 vs 25 pmoles/ml) after one BNP injection (*Figure 1C*). In infarcted mice, 1 day after BNP injection, the plasmatic cGMP concentration remained high (207 vs 51 pmoles/ml, p=0.05) (*Figure 1C*), but returned to the level detected in the plasma of saline-injected mice 2 days after BNP injection (data not shown). The increased cardiac and plasmatic cGMP concentrations demonstrate that injected BNP binds to NPR-A and/or NPR-B receptors.

PLB protein is expressed by contractile (cardiomyocytes and smooth muscle cells) and also by non-contractile cells, such as endothelial cells (*Sutliff et al., 1999*). PLB phosphorylation after BNP injections was evaluated by western blot analysis on proteins extracted from all cardiac tissue or isolated NMCs (*Figure 1D–E*). The pPLB/PLB ratio increased 2.1-fold in 'unmanipulated' hearts 1 to 3 hr after BNP injection (*Figure 1D*). In infarcted hearts of BNP-treated mice, the pPLB/PLB ratio increased 3 days after MI, 2.1- and 1.5-fold in the infarct and border zone (ZI+BZ) and in the remote zone (RZ), respectively. 10 days after MI, the pPLB/PLB ratio increased 4.0-fold in the ZI+BZ and was unchanged in the RZ. Furthermore, the pPLB/PLB ratio increased 197- and 179-fold in NMCs

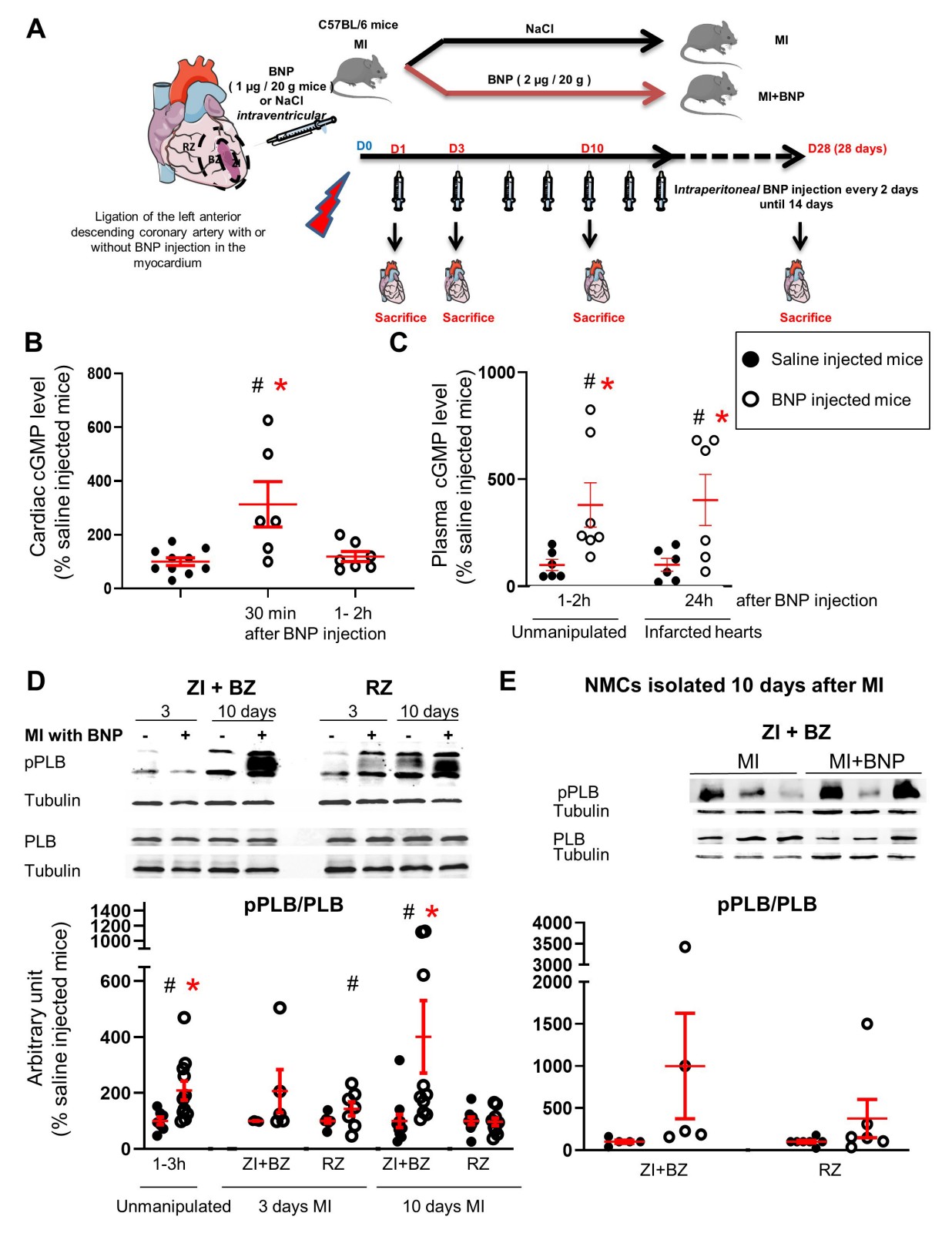

**Figure 1.** Intraperitoneal BNP injection acts on cardiac non-myocyte cells (NMCs). (A) Experimental protocol as described in details in Material and methods section. (B) cGMP level measurement in cardiac tissue of unmanipulated mice injected or not with BNP for 30 min and 1–2 hr. n = at least six mice. (C) cGMP plasma level measurement in unmanipulated or infarcted mice injected or not with BNP. n = 6–7 mice for unmanipulated hearts, n = 6 infarcted mice 24 hr after injection. (D) Representative western blots of total proteins isolated hearts of saline or BNP-injected mice, 3 and 10 days after

*Figure 1 continued on next page*

*Figure 1 continued*

surgery. Blots were stained with antibodies against phospho phospholamban (pPLB), phospholamban (PBL) and Tubulin (used as loading control). Only the bands at the adequate molecular weight were represented here: Tubulin 55 kDa, pPLB between 17 and 26 kDa and PLB 25 kDa. Quantification of the pPLB/PLB ratio. Data obtained from western blot analysis on unmanipulated (n = 7–11 mice per group) and infarcted hearts of mice treated or not with BNP. Results of BNP-treated hearts expressed relatively to the average of saline-treated hearts. 3 days after MI: n = 5 mice for the ZI+BZ and 7–8 mice for the RZ 10 days after MI: n = 10–11 mice for the ZI+BZ and n = 9–10 mice for the RZ. (E) NMCs were isolated from both areas of infarcted hearts treated or not with BNP 10 days after surgery. Proteins were extracted from these cells (n = 5 independent isolation per group for the ZI+BZ and n = 6–7 for the RZ) and pPLB/PLB ratio was evaluated. Only the western blots obtained for NMCs isolated from the ZI+BZ were represented. For B, C, D and E: Individual values are represented and the means ± SEM are represented in red. Statistical analysis was performed only for groups with n ≥ 6. # p<0.05 for different variance between groups, *p<0.05 using unpaired T tests with or without Welch's corrections.

The online version of this article includes the following figure supplement(s) for figure 1:

**Figure supplement 1.** Adult cardiac endothelial cells express BNP receptors in both infarcted and border (ZI+BZ) and remote (RZ) zones.

isolated from the ZI+BZ and RZ of infarcted hearts in BNP-treated mice 10 days after MI (*Figure 1E*). According to these results, intraperitoneal injections of BNP can target cardiac NMCs.

## Increased number of endothelial cells in infarcted hearts

Endothelial cells in infarcted hearts express NPR-A and NPR-B (*Figure 1—figure supplement 1*). NMCs were thus isolated from both areas of infarcted hearts in saline and BNP-treated mice by enzymatic digestion and characterised for genes specific to endothelial cells by quantitative reverse transcription polymerase chain reaction (qRT-PCR). Increased mRNA levels coding for CD31 (x 1.3, p=0.025) and Ve-cadherin (x 1.3, p=0.07) were detected in the cells isolated from RZ of BNP-treated infarcted hearts 3 days after MI (*Figure 2A*). 10 days after MI, mRNA levels coding for vWF (x 1.5, p=0.023), VeCad (x 1.4, p=0.0007) and eNOS (x 1.4, p=0.049) were increased in the ZI+BZ after BNP treatment (*Figure 2A*). The number of CD31+ cells per mg of cardiac tissue was then evaluated in the ZI+BZ and RZ of infarcted hearts by flow cytometry analysis (*Figure 2B–C*). The number of CD31+ cells increased in the RZ (+ 100%) 3 days after MI following BNP treatment (*Figure 2C*). A higher number of CD31+ cells was found 10 days after MI in the ZI+BZ (+29%, p=0.04) and RZ (+28%, p=0.01) of BNP-treated hearts (*Figure 2C*). This was confirmed by western blot analysis (*Figure 2D*). CD31 protein levels were higher in the ZI+BZ (+26%, p=0.06) and RZ (+69%, p=0.0003) of BNP-treated hearts compared to saline-injected hearts 10 days after MI (*Figure 2D*).

Finally, cardiac vascularisation (evaluated by CD31 staining intensity) was determined 3, 10, and 28 days after MI in the BNP- or saline-treated hearts of mice (*Figure 3A–B*). Cardiac vascularisation increased 2.2-fold 3 days after MI in the RZ (p=0.002) of BNP-treated hearts, while it remained unchanged in the ZI+BZ. BNP treatment increased cardiac vascularisation 10 after MI in the ZI+BZ (+ 108%, p=0.02) and RZ (+76%, p=0.002) (*Figure 3A–B*). 4 weeks after MI, vascularisation remained 1.7-fold increased in BNP-treated hearts. We counted CD31+ cells on heart slices after immunostaining (*Figure 3C*), observing a 2.0 and 1.8-fold increase 3 days after MI in the ZI+BZ (p=0.003) and RZ (p=0.024) of BNP-treated hearts compared to saline-injected hearts, respectively. A 1.4- and 2-fold increase in CD31+ cells was counted 10 days after MI in the ZI+BZ (p=0.02) and RZ (p=0.05) of BNP-treated mice, respectively. This was also the case 28 days after MI (ZI+BZ: x 1.8, and RZ: x 2) (*Figure 3C*).

In vitro studies allowed identifying by which receptor BNP acts. NMCs isolated from neonatal or adult hearts were treated or not with BNP until confluence in vitro (*Figure 4*). Cell cultures were analysed by qRT-PCR and the number of CD31+ cells evaluated by flow cytometry analysis. mRNA levels coding for von Willbrand factor, Ve-cadherin, and eNOS were upregulated in neonatal BNP-treated cells compared to untreated cells (*Figure 4A*). BNP treatment increased the number of CD31+ cells in vitro (+87% and +41% in neonatal and adult NMCs, respectively) (*Figure 4C*). The increase in CD31+ cells after BNP treatment completely disappeared in the NMCs isolated from the NPR-A knockout mouse model (*Npr1* KO) but not from NPR-B KO hearts (*Npr2* KO hearts) (+56%, p=0.04), suggesting that BNP binds to NPR-A to increase the number of endothelial cells (*Figure 4C*).

Overall, our results clearly demonstrate that BNP injections after MI lead to more endothelial cells initially in the RZ (3 days after MI) and then in the ZI+BZ (10 days after MI). This is also the case in unmanipulated adult hearts where intraperitoneal BNP injections every 2 days for 10 days increase

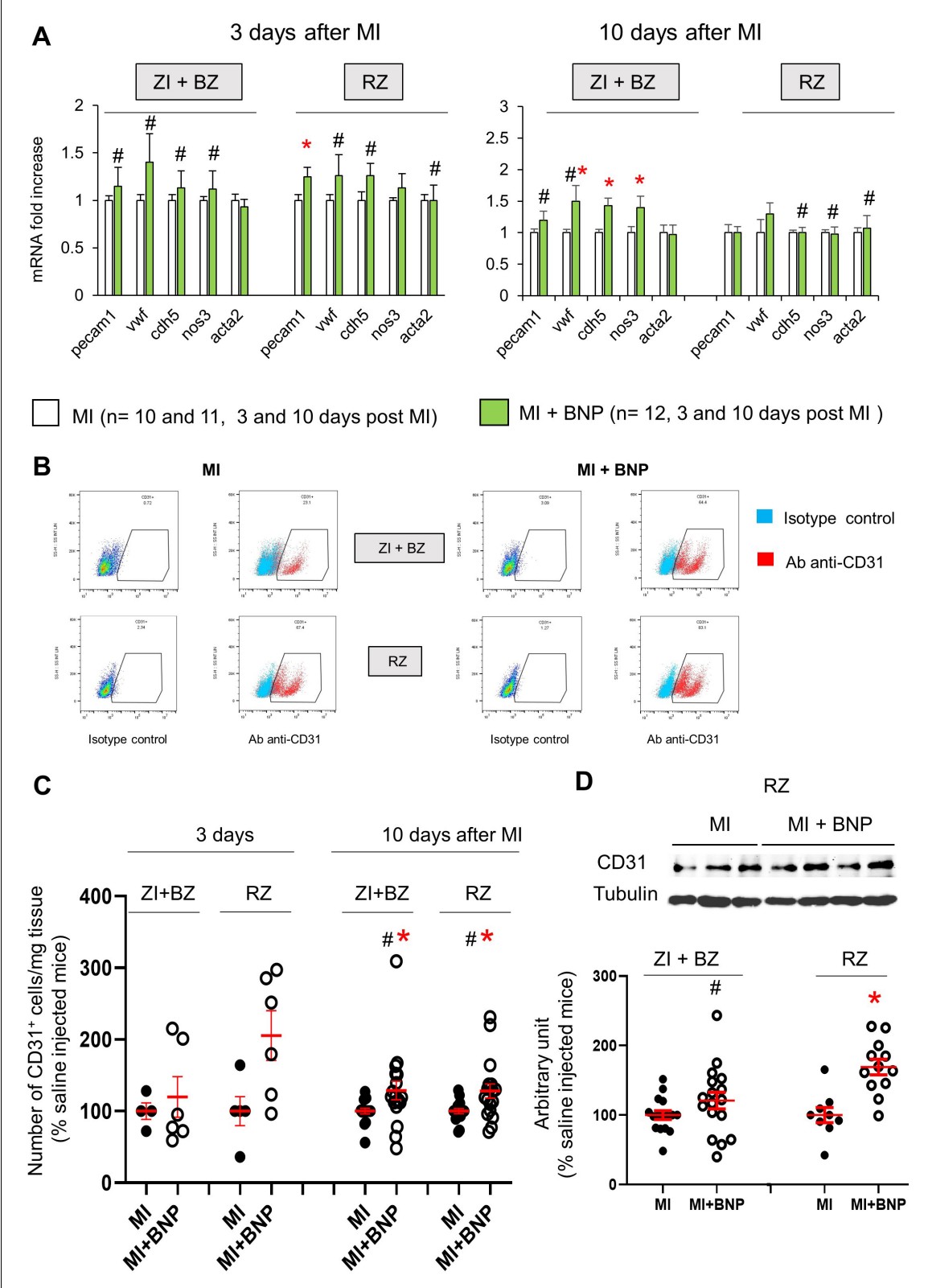

**Figure 2.** Increased endothelial cell number in infarcted hearts after BNP treatment. (**A**) Quantitative relative expression of mRNAs coding for endothelial cell specific proteins (CD31 (*pecam1* gene), von Willbrand factor (*vwf* gene), Ve-cadherin (*cdh5* gene), eNOS (*nos3* gene)), alpha smooth muscle actin (alpha SMA) (*acta2* gene) in the ZI+BZ and RZ areas of saline (MI) and BNP-injected hearts (MI+BNP) 3 and 10 days after surgery. Results expressed as fold-increase above the levels in saline-injected infarcted mice. Results are represented as mean ± SEM. *p<0.05. (**B**) Representative flow

*Figure 2 continued on next page*

*Figure 2 continued*

cytometry analysis of NMCs isolated from the ZI+BZ or RZ of infarcted hearts after BNP or saline treatments 10 days after MI. NMCs stained with control isotype or antibody against CD31 protein. Analysis performed on DAPI negative cells (i.e. living cells). (C) Quantification of the data obtained by flow cytometry analysis on NMCs isolated from infarcted hearts 3 and 10 days after MI. The number of CD31$^+$ cell in BNP-treated hearts related to the number obtained in saline-injected hearts. 3 days after MI: n = 4 MI and 6 MI + BNP mice. 10 days after MI: n = 16 MI and 15 MI +BNP mice. (D) Representative western blot of proteins extracted from the ZI+BZ of MI and MI+BNP hearts 10 days after surgery. Blots were stained with antibodies against CD31 and Tubulin (used as loading control). Only the bands at the adequate molecular weight were represented here: Tubulin (55 kDa), CD31 (130 kDa). Quantification of the data from western blot analysis expressed relative to the average of MI hearts. Results were from n = 15–16 different hearts for the ZI+BZ and n = 9–12 hearts for the RZ. (C, D) Individual values are represented and the means ± SEM are represented in red. Statistical analysis was performed only for groups with n ≥ 6. # p<0.05 for different variance between groups, *p<0.05 using unpaired T tests with or without Welch's corrections.

heart vascularisation (+37%, p=0.0007) (*Figure 3—figure supplement 1*). In vitro experiments suggest that BNP acts via NPR-A.

## Mobilisation of resident mature endothelial cells and precursor cells

The increased number of endothelial cells in infarcted hearts after BNP treatment resulted from either the direct effect of BNP on pre-existing cardiac endothelial cells and/or the effect of BNP on other cells. Indeed, BNP may increase the number of infiltrating endothelial cells or stimulate the differentiation of endothelial precursor cells into endothelial cells in infarcted hearts. We therefore studied the origin of endothelial cells in infarcted hearts of BNP-treated mice.

We first investigated whether the increased number of endothelial cells in BNP-treated hearts originated from infiltrating cells. We performed immunostainings against CD45 and CD31 proteins and we counted CD31$^+$ and CD45$^+$ cells in infarcted hearts treated or not with BNP 3 and 10 days after MI (*Figure 3—figure supplement 2*). The percentage of CD45$^+$ cells among the CD31$^+$ cell subset was less than 10% in all zones of the infarcted hearts 3 and 10 days after MI. The numbers and percentages of CD45$^+$ and CD31$^+$ cells were similar in BNP- and saline-treated hearts 3 and 10 days after MI. The increased number of CD31$^+$ cells in BNP-treated hearts was thus not due to infiltrating cells.

To understand whether the increased number of endothelial cells in BNP-treated hearts originated from pre-existing endothelial cells or from the differentiation of cardiac precursor cells, heterozygous tamoxifen-inducible Cdh5:ROSA26 mice were used to trace CD31$^+$ cells (*Figure 5—figure supplement 1*). Tamoxifen injections given 2 weeks before MI induced green fluorescent protein (GFP) expression in CD31$^+$ cells (*Figure 5—figure supplement 1A–C*). To avoid contamination with GFP$^-$ cells, our analysis focussed on the CD31$^+$ cell subset with 94% expressing the GFP protein before MI (*Figure 5—figure supplement 1C*). Ten days after MI, immunostainings showed that almost all CD31$^+$ cells expressed the GFP protein in the ZI+BZ and RZ of BNP-treated and untreated infarcted hearts (*Figure 5A–B*). As shown in *Figure 5—figure supplement 1D*, more GFP$^+$ cells were apparent in the ZI+BZ of BNP-treated mice compared to those injected with saline. Numerous vessels and capillaries formed by GFP$^+$ cells were detected in this area after BNP injections. However, we also detected some GFP$^-$ endothelial cells in the ZI+BZ of BNP- and saline-treated hearts (*Figure 5B*).

We quantified this observation by isolating NMCs in infarcted Cdh5:ROSA hearts. The percentage of CD31$^+$ cells expressing or not the GFP protein was determined by flow cytometry. BNP treatment increased the number of CD31$^+$ GFP$^+$ cells (i.e. originating from pre-existing endothelial cells) in the ZI+BZ (+37%, p=0.02) and RZ (+52%, p=0.03) (*Figure 5C*) of all treated hearts. Interestingly, the number of CD31$^+$ GFP$^-$ cells increased significantly in the ZI+BZ (+95%, p=0.07) but not in the RZ (+23%, p=0.2) after BNP treatment (*Figure 5C*).

Our results demonstrated that endothelial cells in the infarcted hearts of BNP-treated mice originate mainly from pre-existing endothelial cells in the ZI+BZ and RZ. However, endothelial cells originating from precursor cells (i.e. GFP$^-$) also contribute to the neovascularisation of the ZI+BZ of BNP-treated infarcted hearts.

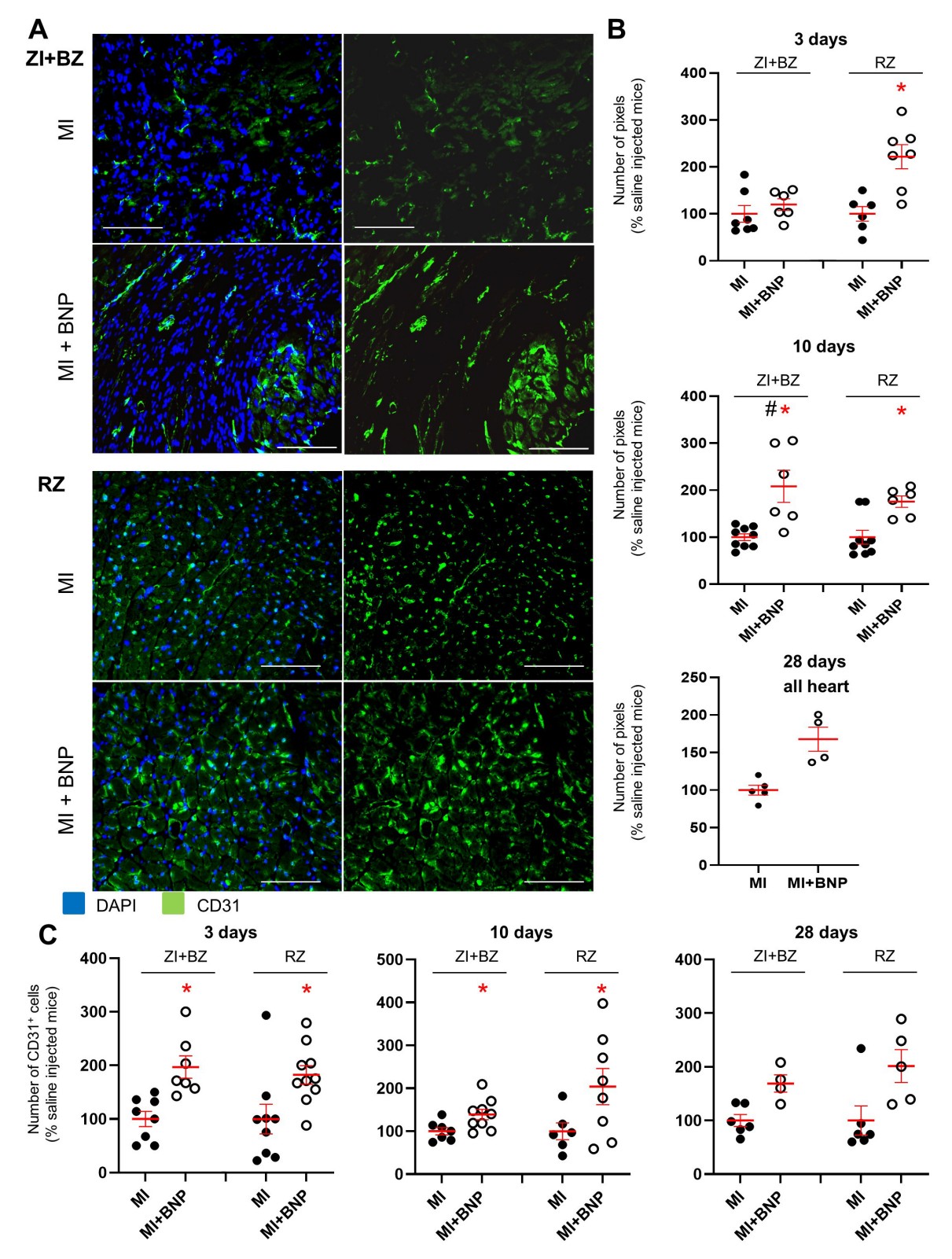

**Figure 3.** Increased vascularisation in BNP-treated infarcted hearts. (**A**) Representative immunostainings against CD31 protein (green) on hearts removed from saline-(MI) and BNP-treated infarcted mice (MI + BNP) 10 days after surgery. Nuclei stained in blue with DAPI. Scale bars: 100 µm. (**B**) CD31 staining intensity measured on at least 10 different pictures per heart and per area 3, 10 and 28 days after MI. Number of pixel in BNP-injected mice related to the numbers of saline-injected mice. (**C**) CD31[+] cell number counted on heart sections of the different area of saline- and BNP-treated

*Figure 3 continued on next page*

*Figure 3 continued*

infarcted hearts. Cells counted on at least 10 different pictures per area and mouse. (B–C): Individual values are represented and the means ± SEM are represented in red. Statistical analysis was performed only for groups with n ≥ 6. # p<0.05 for different variance between groups, *p<0.05 using unpaired T tests with or without Welch's corrections.

The online version of this article includes the following figure supplement(s) for figure 3:

**Figure supplement 1.** BNP injection led to increased vascularisation in unmanipulated hearts.

**Figure supplement 2.** Few cardiac endothelial cells in infarcted and remote zone 3 and 10 days after infarction express the CD45 protein.

## Stimulated proliferation of endothelial cells via p38 MAP kinase activation

We then investigated the capacity of BNP to stimulate the proliferation of endothelial cells (*Figure 6*) by performing immunostaining against CD31 and 5-Bromo-2′-deoxyuridine (BrdU) on BNP- and saline-treated infarcted hearts 1–3 and 10 days after surgery (*Figure 6A–B*). To obtain the percentage of proliferating endothelial cells in each area of the infarcted hearts, the number of CD31$^+$ BrdU$^+$ cells was divided by the total number of CD31$^+$ cells (*Figure 6B*). During the first days after surgery (1–3 day after), BNP stimulated endothelial cell proliferation in the RZ (+53%, p=0.02). In the ZI+BZ, higher endothelial proliferation following BNP treatment was detected only 10 days after MI (+56%, p=0.02) (*Figure 6B*).

To evaluate the direct BNP effect on endothelial cell proliferation, NMCs from adult hearts expressing fluorescent ubiquitination-based cell cycle indicator (FUCCI) were isolated and cultured for 3–4 days with and without BNP (*Figure 4D*). Transgenic FUCCI mice allow the differentiation of cells in different phases of the cell cycle (*Sakaue-Sawano et al., 2008*). We evaluated the number of CD31$^+$ cells in the phases of the cell cycle by flow cytometry after CD31 staining. The number of adult endothelial cells in the G2/M phase of the cell cycle was 2.2 times higher (p=0.04) in BNP-treated compared to untreated NMCs.

To determine whether higher levels of angiogenic factors could be responsible for increased endothelial cell proliferation in BNP-treated hearts, we determined the vascular endothelial growth factor –A (*vegfa*) mRNA levels in NMCs 3 and 10 days after MI by qRT-PCR. *vegfa* mRNA levels were similar in NMCs isolated from saline- and BNP-treated hearts 3 days after MI (*Figure 6C*). However, *vegfa* mRNA levels increased 10 days after MI in NMCs isolated from the ZI+BZ (+69%, p-=0.02) and RZ (+55%, p=0.03) of BNP-treated hearts compared to NMCs isolated from saline-treated hearts (*Figure 6C*).

We determined the signalling pathway activated by BNP on endothelial cells. For this purpose, we sorted GFP$^+$ endothelial cells from the hearts of unmanipulated Cdh5:ROSA mice injected with tamoxifen 2 weeks prior (*Figure 6D*) and then stimulated them for 1.5 hr with BNP in vitro. We extracted proteins from these cells and performed western blot analysis. The pp38/p38 ratio was 2.0-fold higher after BNP stimulation on the sorted pure endothelial cells compared to untreated cells (p=0.026). Interestingly, in vivo, in BNP-treated infarcted Cdh5:ROSA hearts, endothelial cells expressing pp38 were also detected (*Figure 6E*), suggesting that BNP is able to act directly on endothelial cells via p38 MAP kinase activation in vitro but also in vivo.

## More NMCs expressing Wilms' tumour one protein

The number of endothelial cells originating from GFP$^-$ cells increased in the hypoxic area of hearts isolated from BNP-treated mice, which could point to the mobilisation of vascular precursors by BNP treatment (*Figure 5C*). We measured mRNA levels coding for proteins expressed by endothelial precursors in NMCs from the ZI+BZ of BNP-treated or untreated infarcted hearts by qRT-PCR (*Figure 5D*). Three days after MI, mRNA levels coding for Flk1 (x 1.5, p=0.01), Sca-1 (x 1.4, p=0.04) and WT1 (x 1.6, p=0.01) increased in cells isolated from the ZI+BZ of BNP-treated infarcted hearts. mRNA levels coding for c-kit did not differ. In vitro, NMCs isolated from neonatal hearts were stimulated or not with BNP for 7–10 days. mRNA levels coding for c-kit (x 2.1), Flk1 (x 1.9), Sca-1 (x 2), and WT1 (x 1.6) were significantly higher after BNP treatment (*Figure 7—figure supplement 1C*). These results suggest that BNP could act on WT1$^+$ cells in vivo and in vitro. We thus verified whether WT1$^+$ cells express BNP receptors, NPR-A and/or NPR-B by immunostainings (*Figure 7—figure supplement 1A*).

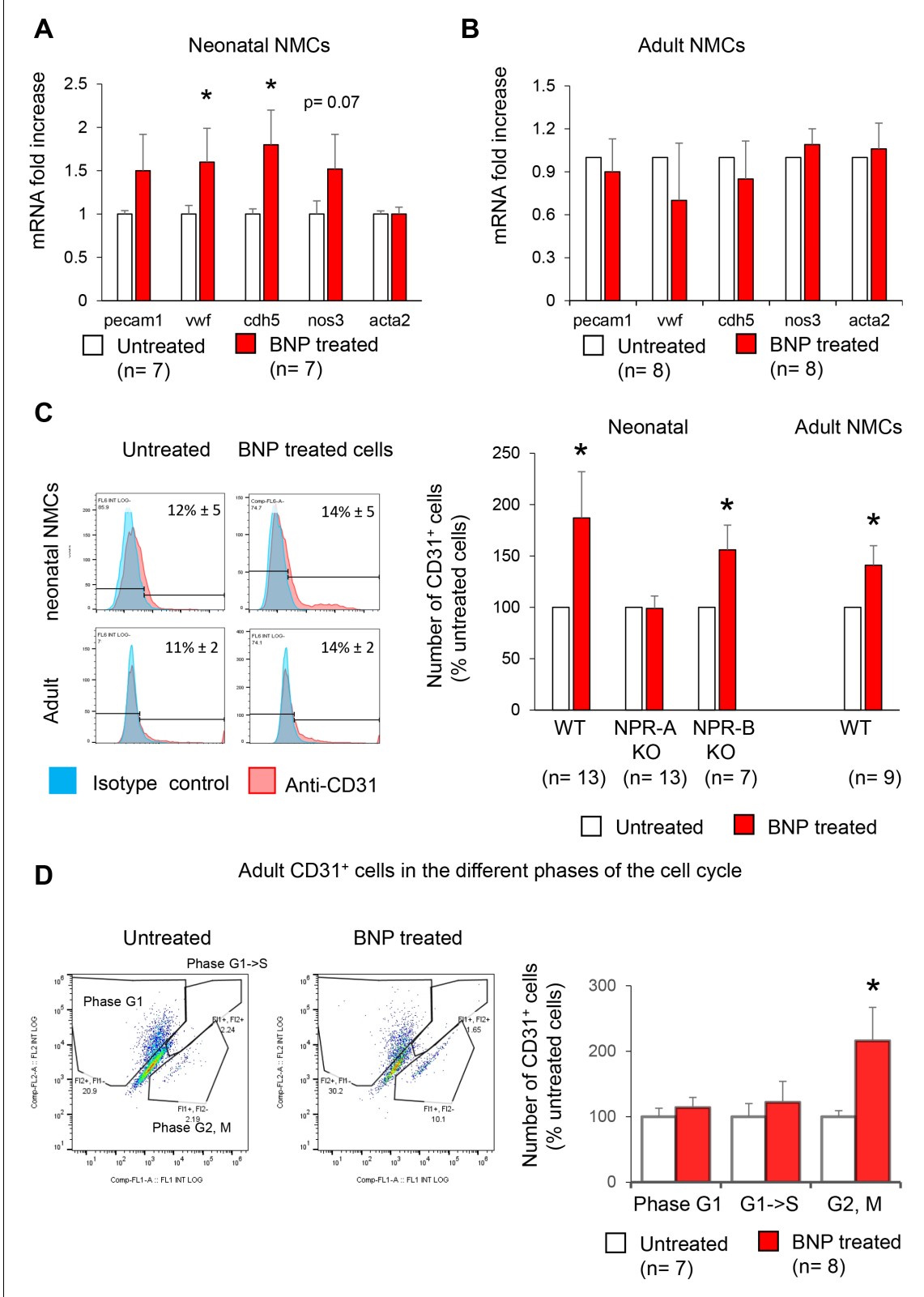

**Figure 4.** Increased number of endothelial cells in vitro after BNP treatment. (**A–B**) Quantitative relative expression of mRNAs coding for endothelial cell specific proteins CD31 (*pecam1* gene), von Willbrand factor (*vwf* gene), Ve-cadherin (*cdh5* gene), eNOS (*nos3* gene), alpha smooth muscle actin (alpha SMA) (*acta2* gene) in NMCs isolated from neonatal (**A**) or adult (**B**) hearts cultured until confluence with or without BNP. Results expressed as fold-increase above the levels in untreated cells. (**C**) Flow cytometry analysis to determine the percentage of CD31+ cells (left: Representative

*Figure 4 continued on next page*

*Figure 4 continued*

histograms) in untreated or BNP-treated NMCs isolated from neonatal or adult hearts. Right: Quantification of the number of CD31$^+$ cells. Results expressed as fold-increase above the number obtained in untreated cells. Neonatal NMCs were isolated from heart of C57BL/6 (WT), NPR-A or NPR-B deficient pups. Adult NMCs were isolated only from C57BL/6 hearts. (D) Adult NMCs isolated from FUCCI mice and treated with or without BNP. Flow cytometry analysis (left: representative dot plots) to determine among the CD31$^+$ cells, the percentages of cells in the G1 phase (Fl2+ Fl1-), in the G1-> S phase (Fl2+Fl1+) and in the G2, M phase (Fl2-Fl1+). Right: Quantification of the number of CD31$^+$ cells in the different phases of the cell cycle. Results expressed as fold-increase above the numbers obtained in untreated cells. For all quantifications, the results are means ± SEM, paired T tests were used, *p<0.05.

We performed also immunostainings on BNP- and saline-treated infarcted hearts to evaluate the number of WT1$^+$ cells (*Figure 7A–C*). WT1$^+$ cells were easily detected in the epicardium and endocardium of adult hearts after MI as reported by others (*Duim et al., 2015*; *Balbi et al., 2019*; *Zhou et al., 2012*). In the ZI+BZ area, compared to saline-injected infarcted hearts, BNP treatment led to more WT1$^+$ cells 3 days (x 2.5 in epicardium and x 3.5 in endocardium) and 10 days after MI (x 2.9 in epicardium and x 1.7 in endocardium) (*Figure 7A–C*). In the RZ of BNP- versus saline-treated hearts, the number of WT1$^+$ cells increased in the epicardium (x 2.5) and in the endocardium (x 2.3) 3 days after MI and in the epicardium 10 days after MI (x 3.6). No difference in the number of WT1$^+$ cells was detected 28 days after MI (data not shown).

## Stimulation of WT1$^+$ cell proliferation near the infarct zone

WT1 is re-expressed by mature endothelial cells after hypoxia (*Duim et al., 2015*). To determine whether BNP stimulates WT1$^+$ cell proliferation and/or WT1 re-expression in endothelial cells, the percentage of proliferating WT1$^+$ cells (number of WT1$^+$ BrdU$^+$ cells relative to the total number of WT1$^+$ cells) was assessed 3 and 10 days after MI in hearts from BNP-treated or untreated mice.

No increased WT1$^+$ cell proliferation was detected in BNP-treated hearts 3 days after MI (data not shown). The proliferation of WT1$^+$ cells only increased in the epicardium and endocardium of the ZI+BZ in BNP-treated infarcted hearts 10 days after surgery (+45%, p=0.03 and +67%, p=0.04, respectively) (*Figure 7D–E*), showing that the higher number of WT1$^+$ cells in other areas of BNP-treated hearts was likely due to WT1 re-expression. Interestingly, in BNP-treated hearts, almost all WT1$^+$ cells localised in the epicardium express BrdU, whereas only 50% of the WT1$^+$ cells proliferate in the endocardium (*Figure 7D*).

Stimulation of WT1$^+$ cell proliferation by BNP treatment was also highlighted in vitro (*Figure 7—figure supplement 1B,D and E*). After BNP stimulation, the number of WT1$^+$ cells increased (+38%, p=0.02) in cultured NMCs isolated from neonatal hearts. Immunostainings against WT1 and BrdU, allowed to demonstrate that BNP treatment stimulated their proliferation (+23%, p=0.003) compared to untreated NMCs.

## Stimulation of WT1$^+$ precursor cell proliferation

The next step was to identify proliferating WT1$^+$ cell origin. In order to discriminate between WT1$^+$ endothelial precursor cells and mature cells re-expressing WT1 after hypoxia, MI was induced in heterozygous inducible WT1:ROSA mice. Three injections of tamoxifen were administered 2 weeks before MI induction. GFP was expressed only in WT1$^+$ cells. Thus, before surgery, 0.4 ± 0.08% of the NMCs were GFP$^+$, while 0.6 ± 0.09% of the CD31$^+$ cells expressed the GFP protein (n = 4 mice).

Ten days after MI, flow cytometry analysis of NMCs isolated from infarcted WT1:ROSA hearts, showed that BNP treatment significantly increased the number of GFP$^+$ cells by 2.3-fold in the ZI+BZ (p=0.05) (*Figure 8A and D*). As shown by immunostainings, GFP$^+$ cells were mainly localised in the epicardium of the ZI+BZ of infarcted hearts (*Figure 8A–B* (left)). However, in the ZI+BZ of BNP-treated hearts, GFP$^+$ cells migrated into the tissue (*Figure 8A–B* (right)), forming vessel-like structures (white arrows in *Figure 8A and C*). This is confirmed by immunostainings against CD31 (*Figure 8B–C*). Indeed, some of the GFP$^+$ cells stained positive for BrdU, Sca-1 (the Stem cell antigen-1 protein), and CD31, showing that WT1$^+$ precursor cells can proliferate and differentiate into endothelial cells, especially after BNP treatment (*Figure 8—figure supplement 1*, *Figure 8B* (right) and 8C). Flow cytometry analysis showed that, 10 days after MI, 10.5 ± 2% of CD31$^+$ cells expressed the GFP protein (versus 5.0 ± 1.3% of endothelial cells in saline-treated hearts (p=0.03)) in the ZI+BZ of BNP-treated hearts. In the RZ, 8.5 ± 2% of endothelial cells originated from GFP$^+$ cells following

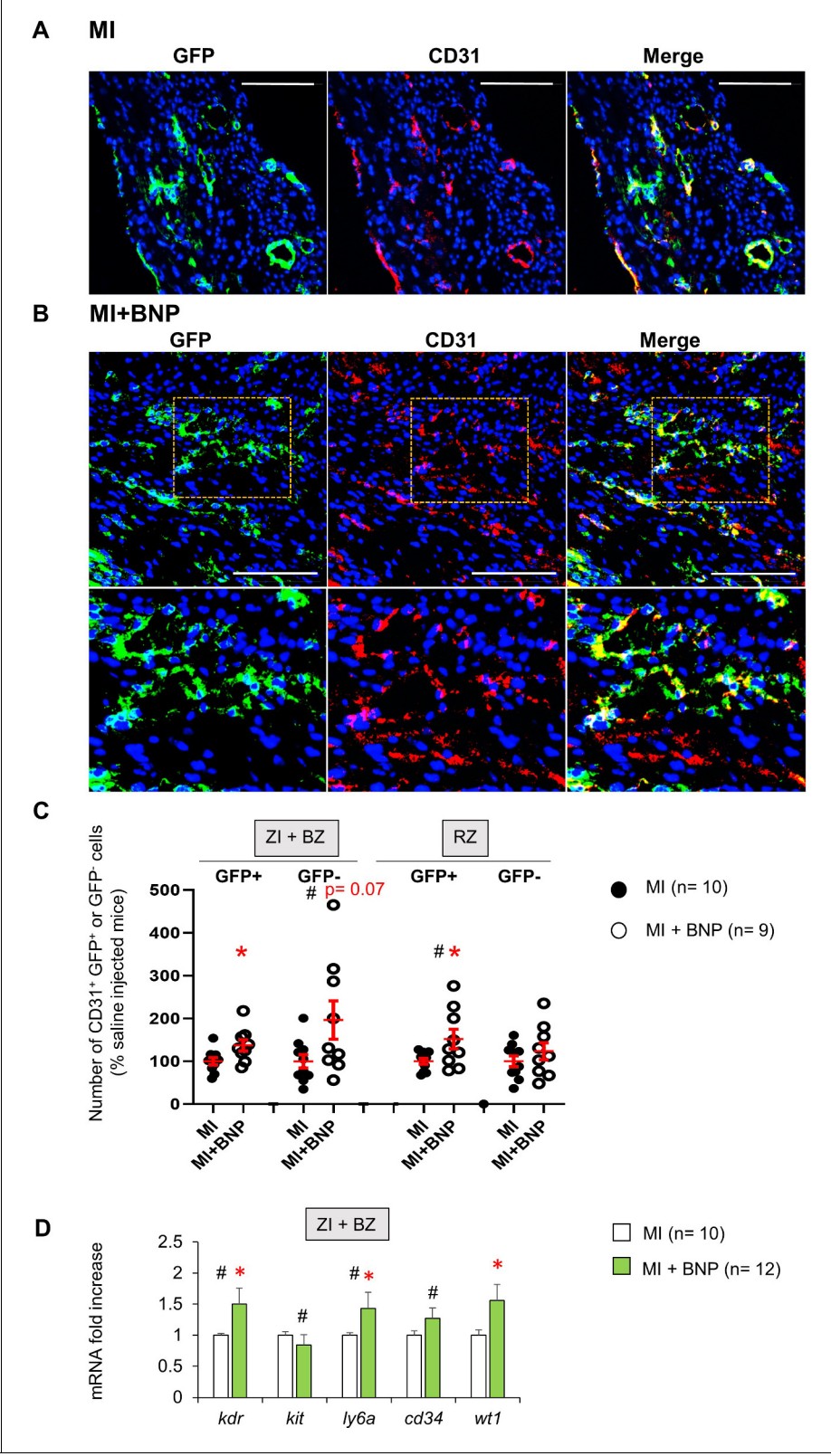

**Figure 5.** Mobilisation of resident mature endothelial and precursor cells in BNP-treated hearts. (**A**) Representative pictures of the ZI+BZ area of Cdh5: ROSA26 infarcted hearts 10 days after surgery, treated (**B**) or not (**A**) with BNP and stained with DAPI (nuclei in blue) and antibody against CD31 protein (red). Endothelial cells or cells originating from CD31[+] cells express GFP protein. Scale bars represent 100 μm. (**B**) Orange rectangles are represented at high magnification below. (**C**) Quantification of the number of GFP[+] and GFP[-] CD31[+] cells. Results expressed in BNP-injected mice as fold-increase

*Figure 5 continued on next page*

Figure 5 continued

above the numbers obtained in saline-injected mice. Individual values are represented and the means ± SEM are represented in red. (D) Quantitative relative expression of mRNAs coding for endothelial precursor specific proteins (Flk-1 (*kdr* gene), c-kit (*kit* gene), stem cell antigen 1 (Sca-1) (*ly6a* gene), CD34 (*cd34* gene) and Wilms' tumour 1 (WT1) (*wt1* gene)) in ZI+BZ 3 days after MI. Results expressed as fold-increase above the levels in saline-injected mice. Results are means ± SEM. (C–D): # p<0.05 for different variance between groups, *p≤0.05 using unpaired T tests with or without Welch's corrections.

The online version of this article includes the following figure supplement(s) for figure 5:

**Figure supplement 1.** Characterisation of the Cdh5:ROSA mouse model.

BNP treatment versus 4.0 ± 1% in saline-treated hearts (p=0.03) (*Figure 8E*). However, among the GFP$^+$ cells, the percentage of cells differentiating into CD31$^+$ cells was the same (around 48%) in the ZI+BZ and RZ between BNP- and saline-treated hearts (*Figure 8F*).

Our results demonstrated that WT1$^+$ precursor cells have the capacity to differentiate into endothelial cells in infarcted hearts. BNP increased the number of endothelial cells originating from WT1$^+$ cells by stimulating WT1$^+$ cell proliferation but not their differentiation into endothelial cells. The signalling pathway by which BNP stimulated WT1$^+$ cell proliferation remains to be identified but we detected phosphorylation of the p38MAP kinase in some GFP$^+$ cells from BNP-treated WT1:ROSA hearts (*Figure 6E*).

## Increased vascularisation in infarcted hearts after LCZ696 treatment

LCZ696 (Entresto, Novartis) product associates both an angiotensin receptor blocker (valsartan) and an inhibitor of neprilysin (NEP, sacubitril). In the large, randomized, double-blind PARADIGM-HF trial, LCZ696 treatment has been shown to promote significant benefits in patients with chronic heart failure, when compared to angiotensin-converting enzyme inhibition (enalapril) (*McMurray et al., 2014*). NEP is an endopeptidase able to degrade several factors including the natriuretic peptides. Thus, treatments of rats, rabbits and humans with NEP inhibitor was shown to increase the blood level of the natriuretic peptides (ANP and BNP) and of cGMP (*Gu et al., 2010*; *Kompa et al., 2018*; *Menendez, 2016*). In the plasma of unmanipulated mice, we determined that cGMP concentration increased 3-fold (138 vs 44.5 pmoles/ml) 24 hr after LCZ696 treatment (60 mg/kg/day). We thus evaluated LCZ696 treatment on heart neovascularisation 10 days after MI. Mice were treated orally by two different concentrations of LCZ696 24 hr after MI induction (*Figure 9A*).

LCZ696 treatment (at the both concentrations) did not affect body weight nor blood pressure (10 days after MI: saline: 103 ± 18 mmHg; LCZ6 treated mice: 103 ± 9 mmHg; LCZ60 treated mice: 109 ± 9 mmHg). Urea and creatinin plasma levels, were not changed after LCZ696 treatment, demonstrating no altered kidney functions (**data not shown**).

LCZ696 treatment prevented the increase of cardiac mass induced by MI (−18%, p=0.005, at the high concentration) (*Figure 9B*). Mice treated with high dose of LCZ696 (60 mg/kg/day) displayed 2.0-fold increased fractional shortening (p=0.02) and 1.9-fold increased ejection fraction (p=0.03) compared to infarcted untreated mice. Moreover, heart remodeling was 2.0-fold decreased during systole and diastole (p=0.03 in systole) (*Figure 9C*). Cardiac functions and parameters didn't change with low concentration of LCZ696 compared with H$_2$0-treated infarcted mice.

Cardiac vascularisation (evaluated by CD31 staining intensity) increased 1.7-fold in both area of LCZ696-treated hearts (at the both concentrations) when compared to untreated infarcted hearts (*Figure 9D*). The proliferations of endothelial and WT1$^+$ cells were evaluated by immunohistochemistry and cell counting (*Figure 9E–G*). LCZ696 treatment stimulated the proliferation of the endothelial cells by 1.8 fold (at both concentrations) in the ZI+BZ and 1.6 and 1.4 fold (for 6 and 60 mg/kg/day, respectively) in the RZ (*Figure 9E–F*). A 1.4 - and 1.5-fold increase in the percentages of proliferating WT1$^+$ cells were determined in the ZI+BZ of LCZ 6 and 60 mg/kg/day-treated hearts, respectively (*Figure 9E–G*). LCZ696 treatment didn't stimulate the WT1$^+$ cell proliferation in the RZ of infarcted-treated hearts.

Our results demonstrated that only the high concentration of LCZ696 was associated with increased heart function and decreased heart remodelling 10 days after MI in mice. However, LCZ696 administration at both concentrations increased cardiac vascularisation in both areas of infarcted hearts. This is likely the consequence of LCZ696-stimulation of endothelial cell proliferation.

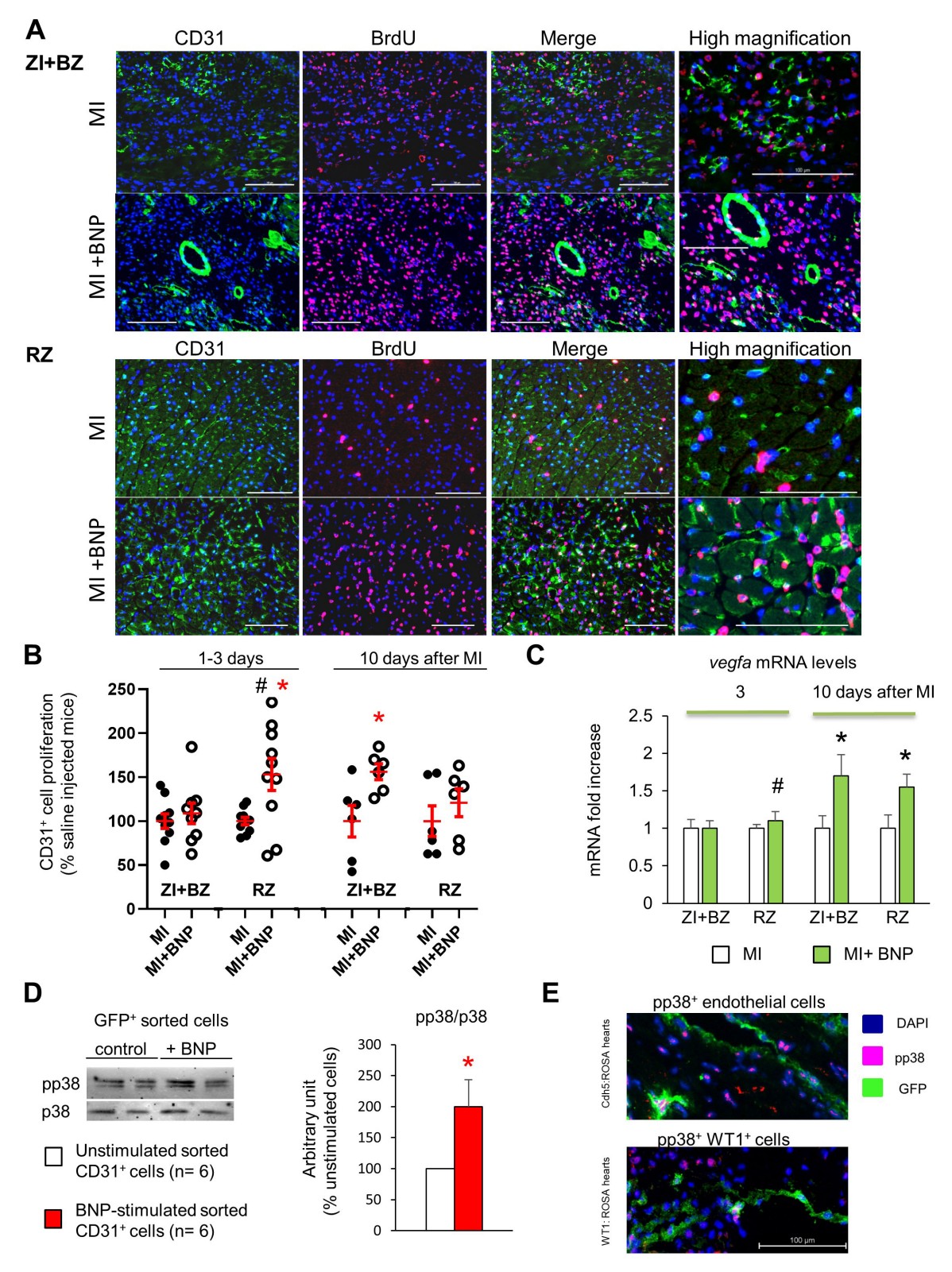

**Figure 6.** BNP stimulation of endothelial cell proliferation. (**A**) Representative pictures of the ZI+BZ and RZ of C57BL/6 infarcted hearts, 10 days after surgery, treated or not with BNP and stained with DAPI (nuclei in blue) and antibodies against CD31 protein (green) and BrdU (pink). Scale bars: 100 μm. (**B**) Percentage of proliferating endothelial cells/per pictures in each area of the infarcted hearts (number of CD31$^+$BrdU$^+$ cells/CD31$^+$ cells). Results in BNP-treated hearts related to those obtained in saline-treated hearts. At least 10 different pictures evaluated per mouse and per area. n = 10–9 mice

*Figure 6 continued on next page*

*Figure 6 continued*

per group 1–3 days after MI, n = 6 mice per group 10 days after MI. Individual values are represented. (**C**) Quantitative relative expression of mRNA coding for VEGF-A in ZI+BZ and RZ 3 and 10 days after MI. Results expressed as fold-increase above the levels in the hearts of saline-injected mice. n = 11–14 hearts per group. (**B–C**) Results are means ± SEM (represented in red). # p<0.05 for different variance between groups, *p≤0.05 using unpaired T tests with or without Welch's corrections. (**D**) NMCs isolated from unmanipulated Cdh5:ROSA26 mice injected 2 weeks before with tamoxifen. GFP[+] cells were sorted and stimulated immediately with or without BNP (5 µg/ml) during 1h30 at room temperature. Western bot analysis was then performed on these cells to evaluate p38 MAP kinase activation. Blots were stained with antibodies against phospho p38 (pp38) (43 kDa), p38 (43 kDa) and Tubulin (55 kDa). Quantification of the pp38/p38 ratio obtained from six independent cell sorting experiments. Results are means ± SEM, *p<0.05 using paired T test. (**E**) Representative pictures of BNP-treated infarcted hearts 10 days after surgery and stained with antibody against pp38. GFP[+] cells represent either endothelial cells (in Cdh5:ROSA mice, picture at the top) or WT1[+] cells (in WT1:ROSA mice, picture at the bottom).

Interestingly, as for BNP-treated hearts, LCZ696 treatment stimulated the proliferation of the WT1[+] cells, but only in the infarcted area.

## Discussion

The work presented here continues our previous research aimed at determining the role of BNP in the heart. We already demonstrated that BNP injections in mice after MI decreased heart remodelling and increased heart function (*Bielmann et al., 2015*). We thus questioned whether increased neovascularisation in infarcted hearts is part of the cardioprotective effect of BNP.

In this study, we first showed that BNP treatment increases myocardial vascularisation and the number of endothelial cells in the infarct zone and border zone (ZI+BZ) as a well as in the remote zone (RZ) of infarcted hearts. Second, BNP stimulates the proliferation of endogenous pre-existing endothelial cells, likely via NPR-A binding and p38 MAP kinase activation. Third, BNP stimulates and/or accelerates the re-expression of the WT1 transcription factor in cardiac cells after MI. Fourth, in the infarcted area of untreated injured hearts, WT1[+] EPDCs proliferate and modestly contribute to heart neovascularisation by differentiating into endothelial cells. Lastly, BNP stimulates WT1[+] EPDC proliferation, with more endothelial cells originating from WT1[+] EPDCs in BNP-treated infarcted hearts.

This is the first work to demonstrate that intraperitoneal injections of BNP increase neovascularisation of the heart after MI. Thus, part of the cardioprotective effect of BNP in infarcted hearts is probably due to accelerated and/or increased neovascularisation in both the ZI+BZ and RZ. This result corroborates works reporting that natriuretic peptides stimulate angiogenesis and vasculogenesis during development and in adult ischaemic organs (*Moyes and Hobbs, 2019*; *Kuhn, 2012*). ANP via NPR-A stimulates endothelial precursor cell proliferation, migration, and differentiation in the skin during cutaneous wound healing (*Lee et al., 2018*). The restoration of blood flow is impaired after hindlimb ischemia in NPR-A KO mice (*Kuhn et al., 2009*), while the injection of a high concentration of CNP increases vascular density in infarcted swine hearts (*Del Ry et al., 2013*). Recently, CNP was identified as a key regulator in angiogenesis and vascular remodelling after ischemia in patients suffering from peripheral artery disease (*Bubb et al., 2019*). Other findings reported that BNP injections in mice lead to increased vascular regeneration in ischaemic limbs (*Shmilovich et al., 2009*) and that intramyocardial BNP gene delivery via adenovirus increased capillary density in normal rat hearts but not in infarcted hearts (*Moilanen et al., 2011*). Our article makes a novel contribution by showing the 'time- and area-dependent' effect of BNP and the identification of both different mechanisms by which BNP increases the number of endothelial cells in infarcted hearts.

In our mouse model, the timing and action mechanisms of BNP stimulation differed in the myocardium in the ZI+BZ and RZ of infarcted hearts. In the RZ of infarcted hearts, the number of endothelial cells and vascularisation increased 3 days after MI. In the ZI+BZ, we detected more endothelial cells 3 days after MI but vascularisation increased only 10 days after MI. BNP stimulates endothelial cell proliferation first in the RZ (1–3 days after MI induction) and then in the ZI+BZ (10 days after MI). Although BNP seems to induce the re-expression of the WT1 transcription factor in cardiac cells in both areas of infarcted hearts (3 to 10 days after MI), its effect on WT1[+] EPDCs is limited to ZI+BZ of infarcted hearts 10 days after MI.

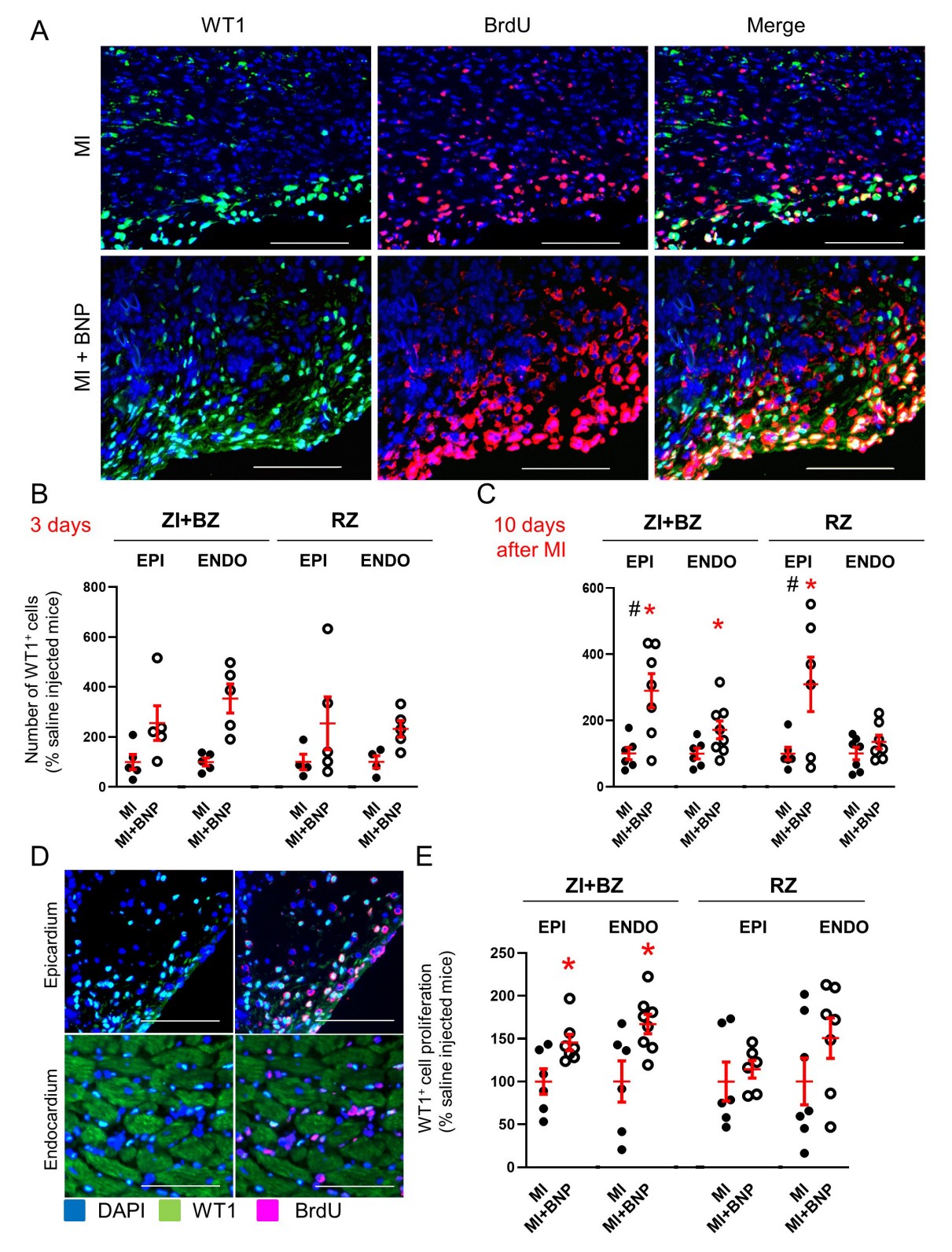

**Figure 7.** Increased number of WT1+ cells in infarcted BNP-treated hearts. (A) Representative pictures of the epicardium of the ZI+BZ area of C57BL/6 infarcted hearts treated or not with BNP 10 days after MI and stained with DAPI (nuclei in blue) and antibodies against WT1 protein (green) and BrdU (red). Scale bars: 100 μm. (B–C) WT1+ cell number per pictures in the ZI+BZ and RZ of infarcted hearts treated or not with BNP 3 (B) and 10 (C) days after surgery. (D:) Representative immunostainings of proliferating WT1+ cells in the epicardium and endocardium of the ZI+BZ area of BNP-treated

*Figure 7 continued on next page*

*Figure 7 continued*

infarcted heart 10 days after MI. Scale bars: 100 μm. (**E**) Percentages of proliferating WT1$^+$ cells (number of WT1$^+$BrdU$^+$ cells/total number of WT1$^+$ cells) 10 days after MI. B, C, E: Results obtained in the epicardium separated from those obtained in the endocardium. Individual values are represented and the means ± SEM are represented in red. # p<0.05 for different variance between groups, *p≤0.05 using unpaired T tests with or without Welch's corrections only for groups with n ≥ 6. EPI: epicardium, ENDO: endocardium.

The online version of this article includes the following figure supplement(s) for figure 7:

**Figure supplement 1.** BNP treatment stimulated WT1$^+$ cell proliferation in vitro.

---

The delay in stimulating endothelial cell proliferation in the different areas of infarcted hearts can be explained by the bioavailability of BNP (less cell deaths, more vessels and capillaries in the RZ), but its effect on WT1$^+$ EPDCs must be favoured by the microenvironment and probably hypoxia. Indeed, in the ZI+BZ, cells undergo a synergistic effect of hypoxia and BNP.

Concerning the mechanisms, we showed that BNP acts directly on mature endothelial cells by stimulating their proliferation. The fact that BNP treatment did not increase the number of endothelial cells in neonatal NMCs isolated from neonatal NPR-A deficient hearts in our study demonstrates that BNP acts via NPR-A on neonatal cardiac cells. In the mouse model of hindlimb ischaemia, proliferating satellite cells secreted BNP, which stimulates angiogenesis in the neighbouring endothelium cells (*Kuhn et al., 2009*). This mechanism is also impaired in NPR-A KO mice, suggesting that regardless of the organ, BNP stimulates mature endothelial cell proliferation via the NPR-A receptor.

In our work, BNP treatment modulated also the fate of immature cells such as WT1$^+$ EPDCs. This is not the first work to report the effect of BNP on immature or precursor cells. BNP level is highly correlated with the number of circulating endothelial precursor cells in patients suffering from heart failure (*Shmilovich et al., 2009*). In vitro, the proliferation, adhesion, and migration capacities of endothelial precursor cells increased in a dose-dependent manner after BNP treatment (*Shmilovich et al., 2009*). Furthermore, BNP stimulated the proliferation of satellite cells in a hindlimb ischemia model, which produced endogenous BNP stimulating the regeneration of endothelial cells (*Kuhn et al., 2009*).

Among cells expressing the WT1 transcription factor, it is important to discriminate between true WT1$^+$ EPDCs and cardiac cells re-expressing the WT1 transcription factor in ischaemic hearts (*Duim et al., 2015*). While the function of WT1 re-expression in infarcted hearts is poorly understood, it seems necessary for endothelial cell proliferation (*Duim et al., 2015*), with our results showing that BNP treatment increases and/or accelerates this process. However, in Cdh5:ROSA infarcted hearts, we detected GFP$^+$ WT1$^+$ BrdU$^+$ cells (i.e. endothelial proliferating cells expressing WT1) as well as GFP$^+$ WT1$^-$ BrdU$^+$ cells (i.e. endothelial proliferating cells without WT1 expression), which shows that WT1 expression is transient or unnecessary for endothelial cell proliferation.

Regarding WT1$^+$ precursor cells, we induced MI in WT1:ROSA hearts and analysed the fate of GFP$^+$ cells. We observed the proliferation of WT1$^+$ EPDCs, mainly localised in the subepicardial layer after MI as previously reported (*Duim et al., 2015*; *Balbi et al., 2019*; *Zhou et al., 2012*). Although the differentiation of WT1$^+$ cells into endothelial cells in infarcted hearts is still debated (*Zhou et al., 2008*; *Zhou et al., 2012*; *Zhou et al., 2011*), we found that 5% of endothelial cells in the ZI+BZ originated from WT1$^+$ EPDCs in saline-treated infarcted hearts. Interestingly, 50% of WT1$^+$ cells differentiated into endothelial cells in these infarcted hearts. This shows that WT1$^+$ EPDCs have the natural capacity to differentiate into endothelial cells in adult hypoxic hearts, as is the case during embryogenesis (*Zhou et al., 2008*). Due their low number, it was nevertheless difficult to highlight this process.

BNP increased the number of WT1$^+$ EPDCs in the subepicardial layer by stimulating their proliferation. Consequently, more cells migrated into the myocardium and differentiated into endothelial cells. We cannot exclude that BNP could also stimulate the migration of EPDCs into the myocardium, where the conditions could be adequate to differentiate into endothelial cells. Regardless of the mechanism, 10% of endothelial cells in the ZI+BZ originate from WT1$^+$ EPDCs in BNP-treated hearts. However, the differentiation capacity of the WT1$^+$ cells into endothelial cells was identical in saline- and BNP-treated hearts (about 50%). BNP therefore stimulates WT1$^+$ EPDC proliferation but not their differentiation into endothelial cells.

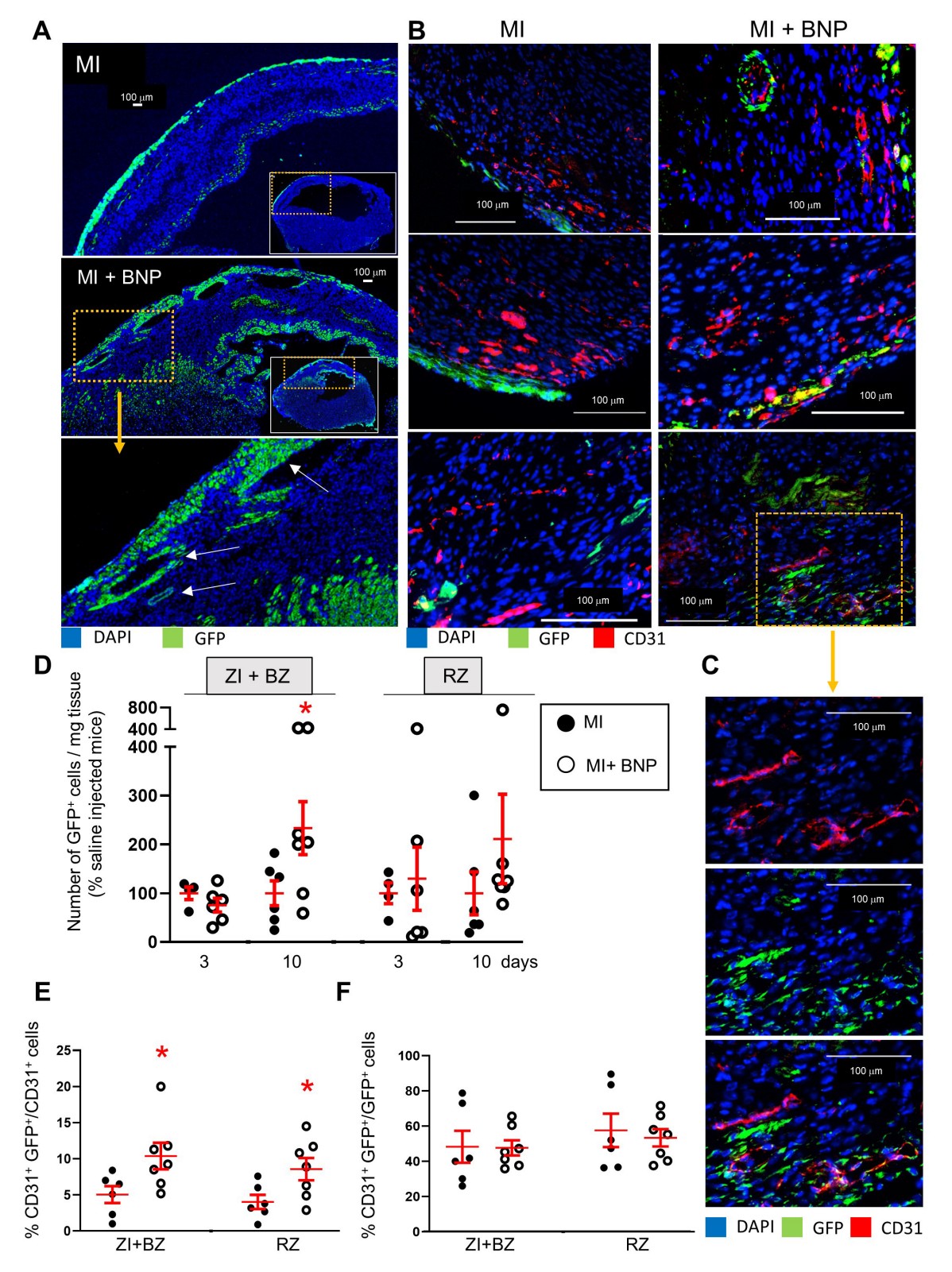

**Figure 8.** Increased WT1[+] cell proliferation after BNP treatment in infarcted hearts. (**A**) Representative immunostainings of ZI+BZ of WT1:ROSA hearts treated or not with BNP 10 days after surgery and stained with DAPI (nuclei in blue) and antibody against GFP protein (green). Hearts represented in full in the small inserts. The orange rectangles delimited the enlarged area below. (**B**) Representative immunostainings of WT1:ROSA hearts treated or not with BNP 10 days after MI and stained with DAPI (nuclei in blue) and antibodies against CD31 protein (red) and GFP (green). White arrows

*Figure 8 continued on next page*

*Figure 8 continued*

represented GFP⁺ CD31⁺ cells, that is endothelial cells originating from WT1⁺ cells. (C) High magnification of a part of the ZI+BZ of infarcted BNP-treated hearts where WT1⁺ cells contributed to the vessel formation (orange rectangle). (D) GFP⁺ cell number per mg of cardiac tissue 3 or 10 days after surgery, determined by flow cytometry analysis. Results in BNP-treated hearts related to those obtained in saline-treated hearts. E and F. Flow cytometry analysis on isolated NMCs stained with antibodies against CD31 and GFP. (E) Percentages of CD31⁺ cells originating from WT1⁺ precursor cells (GFP⁺CD31⁺ cells). The percentages of GFP⁺ cells determined among the selected CD31⁺ cells. (F) Percentages of differentiating WT1⁺ cells into CD31⁺ cells. The percentages of CD31⁺ cells determined among the selected GFP⁺ cells. (D:) 3 days after surgery: MI: n = 4, MI+BNP: n = 6. (D–F:) 10 days after surgery: MI: n = 6, MI+BNP: n = 7 different mice. Individual values are represented and the means ± SEM are represented in red. *p≤0.05 only for groups with n ≥ 6. No difference of variance between groups.

The online version of this article includes the following figure supplement(s) for figure 8:

**Figure supplement 1.** Representative pictures of infarcted WT1:ROSA mice treated or not with BNP, 10 days after surgery.

Our study does not fully elucidate the signalling pathways by which BNP induces endothelial cell and WT1⁺ EPDC proliferation. Additional work is needed in this respect, especially regarding the link between BNP treatment and VEGF expression. Indeed, the reactivation of the VEGFA signalling pathway in adult infarcted hearts 10 days after surgery is likely the key of increased angiogenesis after BNP treatment. This pathway was shown to be inactive in adult ischaemic hearts, whereas it drives angiogenesis after ischemia in neonatal hearts (*Payne et al., 2019*). However, as some studies reported that natriuretic peptides repress VEGF synthesis (*Pedram et al., 2001*), it is unclear which mechanism(s) increased *vegfa* mRNA levels in both areas of the infarcted hearts 10 days after MI.

Interestingly, in our experiment, we did not detect increased *vegfa* mRNA levels in the RZ of infarcted hearts 1 or 3 days after MI, where the proliferation of endothelial cells increased. This suggests that, as shown by *Kuhn et al., 2009* using a hindlimb ischemia model, BNP can also promote endothelial cell proliferation independently of VEGF secretion.

In the ZI+BZ, however, increased *vegfa* mRNA levels seem to induce the stimulation of WT1⁺ cell proliferation. Indeed, by stimulating WT1⁺ EPDCs with BNP, we obtained the same results as Zangi et al., who injected synthetic modified RNA encoding VEGF-A directly into the myocardium of infarcted mouse hearts, which stimulated WT1⁺ cell proliferation and shifted their differentiation into endothelial cells (i.e. 50% of WT1⁺ cells differentiated into endothelial cells) (*Zangi et al., 2013*). Thus, it seems that hypoxia and BNP increase VEGF, which stimulates WT1⁺ EPDC proliferation. The interactions between BNP and VEGF should therefore be studied in more detail to confirm whether BNP acts on endothelial cells via a VEGF-independent mechanism and on WT1⁺ EPDCs via a VEGF-dependent mechanism.

The findings presented in our study hold potential to offer new therapeutic strategies to improve the neovascularisation of hearts after MI or to aid neovascularisation in ischaemic hearts in patients with chronic coronary artery disease. Indeed, we identified BNP as a 'new cardiac angiogenic' factor, which stimulates the proliferation of both resident cardiac mature endothelial cells and WT1⁺ EPDCs. We have not yet identified the exact mechanisms by which BNP could act but re-activation of the VEGF-dependent pathway, active in infarcted neonatal hearts but inactive in adult infarcted hearts, may be possible (*Payne et al., 2019*). We also identified the proliferation and differentiation of WT1⁺ EPDCs as a mechanism, which can be targeted in infarcted adult hearts to increase heart revascularisation. To summarise, BNP binds to NPR-A to stimulate endothelial cell proliferation via p38 MAP kinase activation (*Figure 10*). BNP treatment also stimulates WT1⁺ EPDC proliferation. It remains to be determined whether BNP acts directly on these cells or via increased VEGF level (*Figure 10*). Interestingly, the benefit of LCZ696 treatment which reduces significantly the mortality of patients with chronic heart failure seems also to be associated with increased heart vascularisation, as a result of increased endothelial and WT1⁺ cell proliferation. However, whether this is associated with increased level of BNP remains to be demonstrated.

## Materials and methods

### Mice

All animals were maintained in accordance with the recommendations of the U.S. National Institutes of Health Guide for the Care and Use of Laboratory Animals (National Institutes of Health

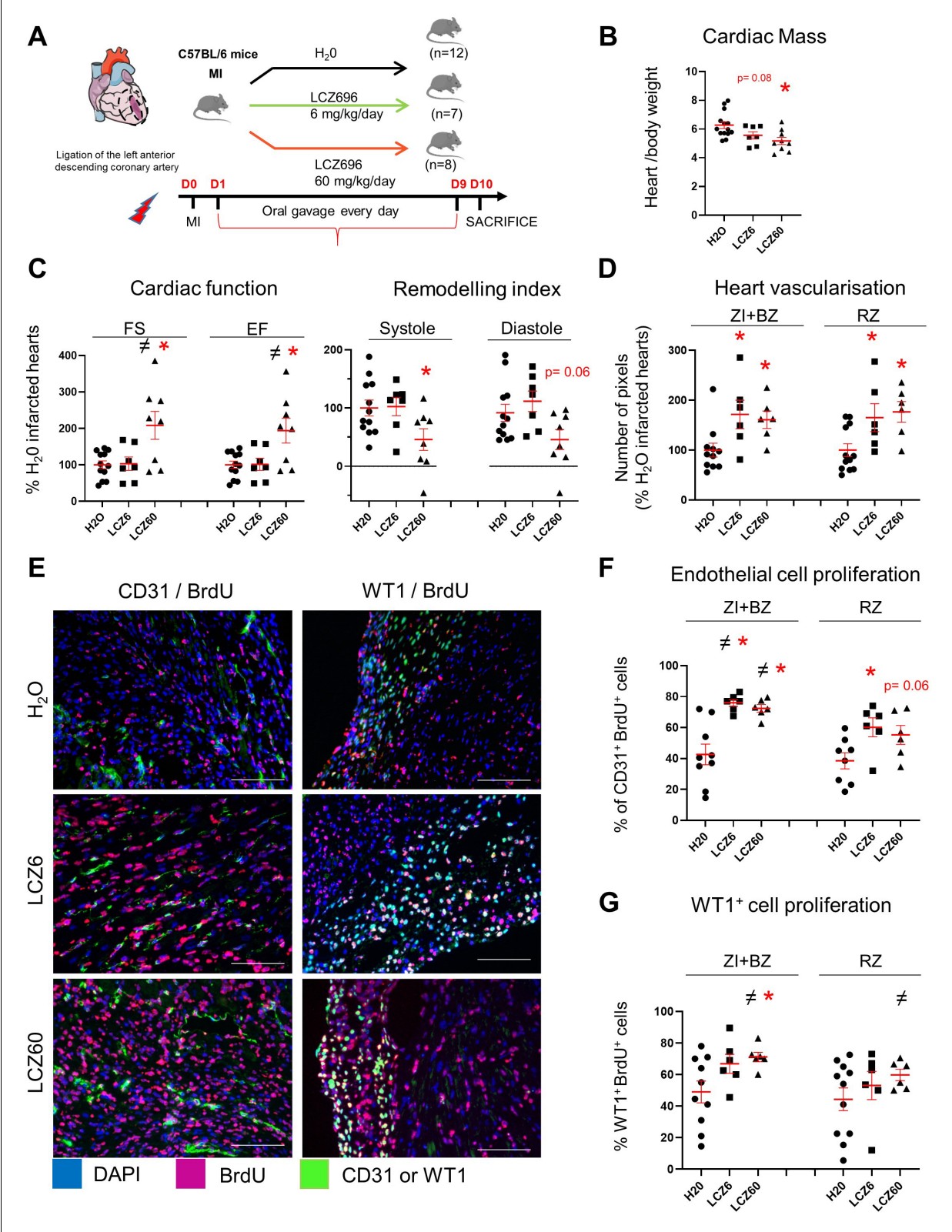

**Figure 9.** Increased vascularisation in infarcted hearts after LCZ696 treatment. (**A**) Experimental protocol as described in details in Material and methods section. (**B**) Cardiac mass (heart weight (mg)/body weight (g)) of infarcted mice 10 days after MI. (**C**) Cardiac function and remodelling index measured by echocardiography 8–9 days after MI (i.e. 1 day before sacrifice). FS: fractional shortening; EF: ejection fraction. Two sets of experiment were performed. All results of the treated mice were related to their respective control (i.e. $H_2O$-treated infarcted hearts). (**D**) CD31 staining intensity

*Figure 9 continued on next page*

*Figure 9 continued*

measured on at least 10 different pictures per heart and per area 10 days after MI. Number of pixels in hearts of LCZ696 treated mice related to the numbers of untreated mice ($H_2O$). (E) Representative pictures of the ZI+BZ area of infarcted hearts 10 days after surgery, treated with LCZ696 (6 or 60 mg/kg/day) or $H_2O$ and stained with DAPI (nuclei in blue) and antibodies against CD31 or WT1 protein (green) and BrdU (pink). Scale bars represent 100 µm. (F–G) Percentage of proliferating endothelial (F) or WT1[+] (G) cells/per pictures in each area of the infarcted hearts (number of CD31[+]BrdU[+] cells/ total number of CD31[+] cells (F) or WT1[+]BrdU[+] cells/total number of WT1[+] cells (G)). At least 10 different pictures evaluated per mouse and per area. B, C, D, F, G: Individual values are represented and the means ± SEM are represented in red. # p<0.05 for different variance between groups, *p≤0.05 using unpaired T tests with or without Welch's corrections.

publication 86–23, 1985). The experiments were approved by the Swiss animal welfare authorities (authorisations VD3111 and VD3292).

C57BL/6 mice (Wild Type mice, WT) were purchased from Janvier (Le Genest-Saint-Isle, France). The NPR-A (−/−) mice (C57BL/*6 Npr1 KO mice*) were kindly provided by Dr Feng Li and Prof Nobuyo Maeda (Chapel Hill, North Carolina, US). The NPR-B deficient mouse strain (*C57BL/6J-Npr2^{slw}*) was generated as heterozygous mice in the Laboratory of Animal Resource Bank at National Institute of Biomedical Innovation (Osaka, Japan) (*Bielmann et al., 2015*; *Rignault-Clerc et al., 2017*). FUCCI (Fluorescence Ubiquitin Cell-Cycle Indicator) mice were provided by Prof Marlene Knobloch (*Sakaue-Sawano et al., 2008*). Inducible (via Tamoxifen) Cre expression under the *Ve*

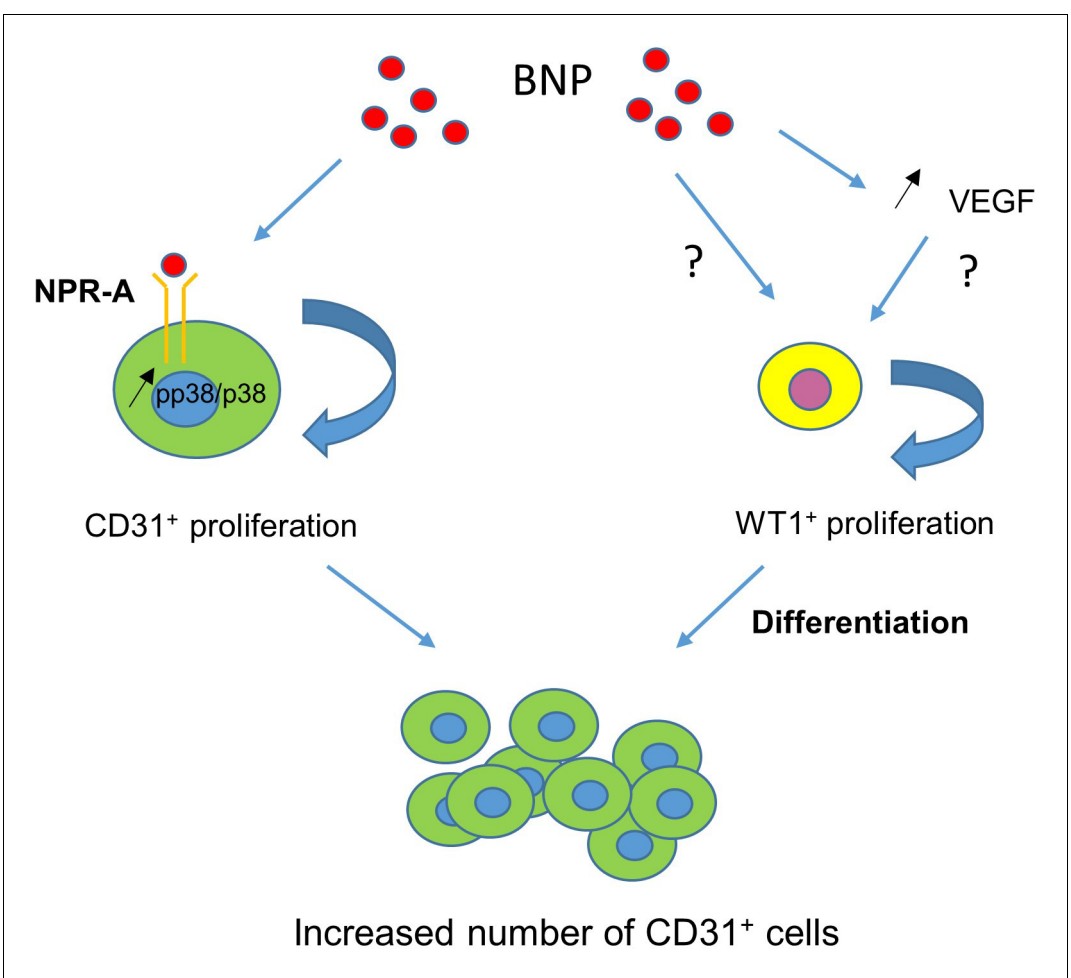

**Figure 10.** BNP-mediated mechanisms leading to increased number of endothelial cells in infarcted hearts. BNP binds directly on NPR-A receptor expressed on endothelial cells and activates p38 MAP kinase to induce their proliferation (left). BNP treatment activates also WT1[+] EPDC proliferation in the ZI+BZ, either directly or via VEGF increase (right).

*cadherin* gene promoter or Cdh5(PAC)-CreER$^{T2}$ mice were provided by Prof Tatiana Petrova (*Monvoisin et al., 2006*). Inducible (via Tamoxifen) Cre expression under the Wilms' tumour 1 homolog gene promoter or *Wt1$^{tm2(cre/ERT2)wtp}$*/J mice were provided by Prof Thierry Pedrazzini (*Rudat and Kispert, 2012*). Gt(ROSA)26Sor$^{tm4(ACTB-tdTomato,-EGFP)Luo}$ mice were purchased from the Jackson Laboratory (JAK 7576). Inducible Cdh5:ROSA26 and Wt1:ROSA26 mice were generated by cross-breeding Cdh5(PAC)-CreERT2 or Wt1$^{tm2(cre/ERT2)Wtp}$/J mice with Gt(ROSA)26Sor$^{tm4(ACTB-tdTomato,-EGFP)Luo}$ mice.

All colonies were established in our animal facility.

## Experimental procedures

Only male mice were used. MI was induced in 8-week-old male C57 BL/6 mice by ligation of the left anterior descending coronary artery (LAD). Briefly, mice were anaesthetised (ketamine (65 mg/kg)/ xylazine (15 mg/kg), acépromazine (2 mg/kg)), intubated and ventilated. The chest cavity was entered through the third intercostal space at the left upper sternal border, and MI was induced by ligature of the LAD with a 7–0 nylon suture at about 1–2 mm from the atria.

For Cdh5:ROSA and Wt1:ROSA mice, surgeries were performed in 8 week-old adult male mice injected 2 weeks before with Tamoxifen.Tamoxifen (Sigma, T5648) was dissolved in ethanol at 100 mg/ml and emulsified in peanut oil to a final concentration of 10 mg/ml. 1 mg Tamoxifen/25 g body weight was injected intraperitoneally (i.p.) to adult mice every 3 days for 3 times (*Bielmann et al., 2015*).

Directly after the surgery, NaCl or BNP (1 μg / 20 g mouse in 20 μl, Bachem synthetic mouse BNP (1-45) peptide (catalog number H-7558)) was injected into the left ventricle cavity. Temgesic (Buprenorphine, 0.1 mg/kg) was injected subcutaneously as soon as the mice waked up and every 8–12 hr during 2 days.

BNP (2 μg / 25 g mouse) was injected i.p. every 2 days (*Bielmann et al., 2015*). After surgery (for the mice sacrificed after 1 day) or 24 hr after the surgery (for all other mice), BrdU (1 mg/ml, Sigma B5002) was added to drinking water and changed every 2 days during 10 days.

For the experiments related to LCZ696 (Entresto, Novartis) treatment, mice after MI were randomly assigned into three different groups: H2O, LCZ696 [6 mg/kg/day], or LCZ696 [60 mg/kg/day]. These two drug concentrations were chosen as LCZ696 [6 mg/kg/day] is the dose mostly used in patients (200–600 mg/day) and LCZ696 [60 mg/kg/day] induces a dose-dependent increase in plasmatic natriuretic peptides in animals (*Suematsu et al., 2016*). LCZ696 drugs were grounded, formulated in water and sonicated for 1 hr before administration. Drugs were administrated 24 hr after MI and once daily for 10 days by oral gavage. Blood pressure was measured daily, from one week before surgery until sacrifice using a tail-cuff based CODA high throughput system (Kent Scientific Corporation).

Mice were sacrificed 1, 3, 10, or 28 days after infarct induction and hearts were removed (*Figure 1A*). If immunofluorescence has to be carried out, apex was embedded into OCT and slowly frozen. Remaining heart was separated into three zones, the infarct zone (ZI), the border zone (BZ) and the remote zone (RZ). According the required experiment, ZI and BZ may be pooled. Tissues were either digested for flow cytometry analysis or quickly frozen for mRNA or protein analysis.

## Cell culture

NMCs were isolated from the hearts of neonatal C57BL/6, NPR-A KO or NPR-B KO pups (1–2 days) as previously described (*Bielmann et al., 2015*; *Rignault-Clerc et al., 2017*) and were cultured in medium composed of MEM Alpha (Gibco 32571–028), 10% FBS, 100 U/ml penicillin G, 100 μg/ml streptomycin with or without BNP (5 μg/ml) up to confluence (i.e. 10–11 days). Adult cardiac NMCs were isolated from adult C56BL/6 mice (6–8 weeks old) by digesting adult ventricles in buffer containing 1 mg/ml collagenase IV (Gibco 17104–019) and 1.2 mg/ml dispase II (Sigma, D4693) and were cultured in EGM−2 Endothelial Cell Growth Medium-2 BulletKit (Lonza, CC-3162) supplemented with 15% foetal calf serum (FCS) (invitrogen Corp) with or without BNP (5 μg/ml) up to confluence (i.e. 5–10 days). Neonatal and adult NMCs were maintained at 37°C in 5% $CO_2$ and 3% $O_2$.

## Endothelial cell sorting

NMCs were isolated from Cdh5:ROSA hearts, injected with Tamoxifen 2 weeks before. GFP$^+$ cells were sorted with the MoFlo Astrios Flow Cytometer System (Beckman Coulter). Then cells were split in half and treated or not with BNP (5 µg/ml) for 1h30 at room temperature. Cells were lysed and proteins extracted.

## Flow cytometry analysis

Cultured neonatal or adult NMCs were removed from dishes using Cell dissociation buffer enzyme-free PBS-based (Gibco 13151–014) and washed in PBS with 3% FCS.

Adult NMCs were isolated from adult infarcted hearts as described above. Samples were treated 5 min at room temperature using CD16/CD32 antibody (BD Biosciences, 553142, 1 µl/10$^6$ cells) and stained with different antibodies listed in *Supplementary file 1*(Supplemental Informations). In case of WT1:ROSA26 cells, cytoplasmic staining was done for GFP detection after fixation (1.5% PFA) and permeabilisation (0.2% saponin). All stainings were performed 20 min on ice. Cells were analysed with Gallios cytometer and data using FlowJo 10 software.

The numbers of CD31$^+$ cells in cell cultures or in NMCs isolated from hearts were obtained by relating the percentage of the CD31$^+$ cells obtained by flow cytometry analysis and the total number of NMCs in culture or obtained after heart digestion in the different area of infarcted hearts. The number of GFP$^+$ or GFP$^-$ cells among NMCs isolated from infarcted Cdh5:ROSA mice was determined by the same method, using flow cytometry analysis.

## Echocardiography and measurements

Transthoracic echocardiographies were performed on adult unmanipulated or infarcted mice using a 30 M-Hz probe and the Vevo 770 Ultrasound machine (VisualSonics, Toronto, Ontario, Canada) as described (*Bielmann et al., 2015*). All measurements were done from leading edge to leading edge according to the American Society of Echocardiography guidelines. Ejection fraction (EF) and fractional shortening, were evaluated on lightly anaesthetised mice (1% isoflurane). Furthermore, according to the fact that changes in left ventricle volume can be considered as an index of remodeling (*Konstam et al., 2011*), we calculated the percentage of increase of the left ventricle volume 10 days after surgery, which is the ratio between (LV Vol;d 1 or 4 weeks − LV Vol;d before surgery) and LV Vol;d before surgery ×100.

## Immunofluorescence

Neonatal and adult hearts were embedded in OCT. Immunostainings were performed on 5 µm heart sections or on cells cultured for up to 11 days on coverslips. Tissue sections or cells were fixed 10 min in 2% PFA. The first antibodies were all incubated overnight at 4°C. Secondary antibodies were incubated 1 hr at room temperature (*Supplementary file 1*). For BrdU detection, heart slides were fixed 10 min in 2% PFA, DNA was denaturated 1 hr at room temperature in HCl 2N before neutralisation in Na Borate 0.1M pH = 8.5, 2 × 5 min. Rat anti-BrdU (1/100, Abcam) was incubated 1 hr at room temperature. Donkey anti-rat was used as secondary antibody. Nuclei were stained with DAPI (0.3 µM). All slides were mounted with Dabco mounting medium (Sigma D2, 780–2) and examined with a Nikon eclipse 90i microscope or Nikon SMZ 25 Stereomicroscope (for the hearts in full, *Figure 8*).

To note, no GFP staining was required to detect the GFP positive cells in the Tamoxifen injected Cdh5:ROSA mice. However, we used an antibody anti-GFP to detect the GFP$^+$ cells in the WT1: ROSA mice injected with Tamoxifen.

To study heart vascularisation, the number of pixels was obtained by processing immunostaining pictures with Adobe Photoshop software.

The percentages of proliferating endothelial (CD31$^+$) or WT1$^+$ cells were obtained by dividing the number of CD31$^+$ BrdU$^+$ cells or WT1$^+$BrdU$^+$ cells obtained by counting per the total number of CD31$^+$ cells or WT1$^+$ cells, respectively.

## Quantitative RT-PCR

Total RNA was isolated from heart tissue or cell culture using Trizol (Ambion 15596026). Reverse transcriptase was carried out using PrimeScript RT Reagent kit with gDNA eraser (perfect Real Time) (Takara, RR047A).

Quantitative real time polymerase chain reaction was performed in duplicates using the TB Green Premix Ex Taq kit (Takara RR420L) on a ViiA 7 Instrument (Applied Biosystems). Results were obtained after 40 cycles of a thermal step protocol consisting of 95℃ (1 s), 60℃ (20 s). The primer sequences were reported in *Supplementary file 2*. Gene expressions were normalized using the housekeeping gene 18S ($\Delta$CT values). Means of $\Delta\Delta$CT values (versus untreated cells) were calculated and results were represented as $2^{-\Delta\Delta CT}$. Statistics were performed on $\Delta\Delta$CT individual values (*Moilanen et al., 2011*).

## Western blot

Total proteins were extracted from tissues or cells as described  and transferred to nitrocellulose membranes before incubation with primary antibodies overnight at 4℃ (*Supplementary file 1*). Secondary antibodies were added 2 hr at room temperature. The immunoblot signals were detected and quantified using the Odyssey infrared imaging system (LI-COR Biosciences, Bad Homburg, Germany). All results were related to their expression of tubulin.

## Determination of cGMP concentration in plasma

cGMP level was detected using the cGMP Enzyme Immuno Assay kit Direct (Sigma). BNP was injected in unmanipulated or infarcted mice. Blood was collected 1–2 hr after BNP injection for unmanipulated mice and 1or 3 days after surgery for infarcted hearts. EDTA-plasma were then processed as recommended in the kit.

## Statistical analysis

All results were presented as mean ± SEM. Statistical analyses were performed only if the number of experiments or mice is ≥6 per group. Paired or unpaired Student-T test were used (*$p < 0.05$). We compared variances between both groups using the F test. If variance is different (#$p < 0.05$), unpaired T test with Welch's correction was used. The alpha level of all tests was 0.05.

## Acknowledgements

The authors thank Dre Corinne Bertonneche, Dr Alexandre Sarre and Ms Anne-Catherine Clerc for her technical expertise. This research was funded by the Swiss National Foundation (PMPDB 310030_162985), the Novartis Foundation for medical-biological research and the Emma Muschamp Foundation (Lausanne).

## Additional information

### Funding

| Funder | Grant reference number | Author |
| --- | --- | --- |
| Schweizerischer Nationalfonds zur Förderung der Wissenschaftlichen Forschung | PMPDB310030_162985 | Nathalie Rosenblatt |
| Emma Muschamp Foundation | F34567 | Nathalie Rosenblatt |
| Novartis Stiftung für Medizinisch-Biologische Forschung | 17B079 | Nathalie Rosenblatt |

The funders had no role in study design, data collection and interpretation, or the decision to submit the work for publication.

## Author contributions

Na Li, Software, Formal analysis, Project administration; Stephanie Rignault-Clerc, Software, Formal analysis, Project administration, perform animal experiments, flow cytometry analysis, molecular biology; Christelle Bielmann, Software, Formal analysis, Visualization, Methodology, perform immunohistology stainings, western blot analysis; Anne-Charlotte Bon-Mathier, Formal analysis, Methodology; Tamara Déglise, Software, Formal analysis, Methodology; Alexia Carboni, master student performing LCZ696 experiments; Mégane Ducrest, Formal analysis, medical student performing immunostainings on endothelial cells on LCZ696 treated mice; Nathalie Rosenblatt-Velin, Conceptualization, Data curation, Software, Formal analysis, Supervision, Funding acquisition, Validation, Investigation, Methodology, Writing - original draft, Project administration, Writing - review and editing

## Author ORCIDs

Nathalie Rosenblatt-Velin  https://orcid.org/0000-0003-2011-2862

## Ethics

Animal experimentation: This study was performed in strict accordance with the recommendations in the Guide for the Care and Use of Laboratory Animals of the National Institutes of Health. The experiments were approved by the Swiss animal welfare authorities (authorisations VD3111 and VD3292). All surgery was performed under sodium pentobarbital anesthesia, and every effort was made to minimize suffering.

## Decision letter and Author response

Decision letter https://doi.org/10.7554/eLife.61050.sa1
Author response https://doi.org/10.7554/eLife.61050.sa2

# Additional files

## Supplementary files

- Supplementary file 1. Antibodies used in flow cytometry analysis, immunohistology and Western blot analysis.
- Supplementary file 2. Primer Sequences used in quantitative RT-PCR.
- Transparent reporting form

## Data availability

All data generated or analysed during this study are included in the manuscript and supporting files.

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
