## [Decision Letter]

**Acceptance summary:**

This manuscript presents data of myocardial angiogenesis in mice subjected to experimental myocardial infarction. The study demonstrates that repeated intraperitoneal injections of synthetic BNP or oral treatment with a drug inhibiting neprilysin-mediated degradation of the endogenous natriuretic peptides, possibly improves cardiac vascularization after myocardial infarction.

**Decision letter after peer review:**

Thank you for submitting your article "Increasing heart vascularisation using brain natriuretic peptide stimulation of endothelial and WT1 + epicardial cells" for consideration by *eLife*. Your article has been reviewed by three peer reviewers, one of whom is a member of our Board of Reviewing Editors, and the evaluation has been overseen by Matthias Barton as the Senior Editor. The reviewers have opted to remain anonymous.

The reviewers have discussed the reviews with one another and the Reviewing Editor has drafted this decision to help you prepare a revised submission.

Summary:

The manuscript entitled "Increasing heart vascularisation using brain natriuretic peptide stimulation of endothelial and WT1 + epicardial cells" by Li et al. reports data of myocardial angiogenesis in mice subjected to experimental myocardial infarction. The study indicates that repeated intraperitoneal injections of synthetic BNP or oral treatment with Entresto, a drug inhibiting neprilysin-mediated degradation of the endogenous natriuretic peptides, possibly improves cardiac vascularization after ischemia.

Microvascular dysfunction after acute myocardial infarction (MI) is a major clinical problem. Although primary percutaneous coronary intervention (PCI) has markedly improved patients' survival, despite epicardial reperfusion more than 30% of patients show signs of microvascular dysfunction leading to adverse left ventricular remodeling and heart failure. Impaired angiogenesis can contribute to myocardial tissue damage. Based on experimental studies, several clinical trials aimed to improve myocardial angiogenesis via intracoronary administration of vascular growth factors, gene transfer or bone marrow mononuclear cells, in patients who had successful primary PCI, but the results were disappointing. A better knowledge of the cellular pathways regulating myocardial (re)perfusion after ischemia is necessary to search for therapeutic strategies capable to restore the microvascular network and flow. The here presented study aimed to elucidate whether B-type natriuretic peptide (BNP) can improve myocardial postischemic angiogenesis in as well as the potential pharmacological, therapeutic implications. Hence, this experimental, “preclinical" study addresses an important, clinically relevant question.

Overall this study follows a very original question. But it includes many different data sets in somehow incomplete way, many of them generated with “NMC". It would profit a lot by concentrating in a clean way on some concrete aspects. Mechanistic studies should be preferentially conducted with sorted or cultured endothelia instead of a mixed cell population (NMC, containing fibroblasts, pericytes, inflammatory cells, besides endothelial cells).

Essential revisions:

1) Certain parts of the study should be completed. For example, why don't the authors present a fine and extensive analysis of cardiac function in animals treated with BNP? In the same way, the authors should complement their experimental approaches with an analysis of all parameters of cardiac remodeling and in particular infarct size and interstitial fibrosis.

2) Conversely, the authors made the effort to analyze cardiac function in animals treated with LCZ696 (Figure 9). However, there is no statistical analysis of these data? or the differences are not significant? in this case, what is the interest of a treatment that increases capillary density without modifying cardiac function? It is however likely that an analysis of cardiac function beyond 10 days post-MI could give significantly different results.

3) The authors should analyze whether or not LCZ696 directly stimulates the proliferation of resident mature endothelial cells and/or that of WT1+ cells.

4) Results: the authors state that "first they determined whether ip BNP acted directly or indirectly on cardiac cells". But there is no single data set in this manuscript allowing to conclude that the observed effects are directly derived from endothelial actions of BNP. As they mention before, BNP acts on many types of cells and organs, and the observed effects could also be "indirect".

5) Results paragraph three: plasma cGMP levels are a poor index of cardiac actions of BNP. It would be more meaningful to measure cardiac cGMP levels.

6) Results paragraph four: it is strange to use the phosphorylation of phospholamban (PLB) as index of BNP activity. This manuscript focuses on angiogenesis. PLB is a regulatory protein in cardiomyocytes. Where is the link to endothelial regeneration?

7) Results paragraph four: BNP increased phosphorylation of PLB by nearly 200-fold in "non-myocyte cells" from the heart. Which cells are these? Is this fraction contaminated by cardiomyocytes? Which non-myocytes have such high PLB levels?

8) How were BNP plasma levels in BNP versus vehicle treated mice after MI? Did Entresto increase BNP plasma levels and to what extend?

9) Most in vitro and ex vivo studies were performed with NMCs. How many endothelial cells are contained in such heterogenous populations?

10) Some basic parameters are missing: how did BNP administration affect cardiac contractile functions as well as the infarct area and area at risk? Did exogenous BNP lower arterial blood pressure?

11) Subsection “Stimulated proliferation of endothelial cells via p38 MAP kinase activation”: how does BNP, via NPR-A/cGMP-signaling, increase MAPK pp38? What is the signaling pathway and do the authors have any hint that this signaling pathway was also activated by BNP in vivo (in endothelial cells in situ)?

12) Subsection "Increased vascularization in infarcted hearts after LCZ696 treatment". But in the corresponding Figure 9, there is no single data set showing “statistically significant effects of entresto”. The figure just shows some preliminary data and trends obtained with very few mice.

13) Figure 1C: it is surprising that the basal levels of pPLB were so low (-). Normally, after MI in mice the endogenous ventricular expression levels of ANP and BNP significantly raise. Was there a difference in pPLB between sham and MI mice (vehicle treatments)?

14) Figure 1D: which types of non-myocyte cells express such high pPLB levels and what is the functional meaning?

[Editors' note: further revisions were suggested prior to acceptance, as described below.]

Thank you for resubmitting your work entitled "Increasing heart vascularisation using brain natriuretic peptide stimulation of endothelial and WT1 + epicardial cells" for further consideration by *eLife*. Your revised article has been evaluated by Matthias Barton (Senior Editor) and a Reviewing Editor.

The manuscript has been much improved but there are some remaining issues that need to be addressed before acceptance, as outlined below:

The current title of the manuscript "Increasing heart vascularisation using brain natriuretic peptide stimulation of endothelial and WT1+ epicardial cells" has been changed from the version originally submitted ("Increasing heart vascularisation after myocardial infarction using brain natriuretic peptide stimulation of endothelial and WT1+ epicardium- derived cells.").

It is important to clarify in the title (and the Abstract) that these findings apply to infarcted but not to healthy hearts, i.e. to post-infarct ischemic injury. The current title is no longer accurate as it does not mention the fact that the observations apply to cells in heart after myocardial infarction / injury. Thus, the title should be modified to "Increasing heart vascularisation after myocardial infarction using brain natriuretic peptide stimulation of endothelial and WT1+ epicardial cells ".

---

## [Author Response]

Essential revisions:1) Certain parts of the study should be completed. For example, why don't the authors present a fine and extensive analysis of cardiac function in animals treated with BNP? In the same way, the authors should complement their experimental approaches with an analysis of all parameters of cardiac remodeling and in particular infarct size and interstitial fibrosis.

The analysis of cardiac function in mice treated or not with BNP was already done and published in our first article (see Figure 7, Bielmann et al., 2014, Basic Research in Cardiology) ^2^. Cardiac structures and functions were evaluated by echocardiography in unmanipulated as well as 1 and 4 weeks after myocardial infarction in saline or BNP-treated mice.

However, we completed here the data and added the results concerning the mRNA level coding for vimentin (a key protein involved in the development of pathological fibrosis) and the estimation of the infarct size obtained by echocardiography. Thus, in the manuscript (Results section), we detailed the results already published and added the new information about vimentin mRNA expression and infarct size.

2) Conversely, the authors made the effort to analyze cardiac function in animals treated with LCZ696 (Figure 9). However, there is no statistical analysis of these data? or the differences are not significant? in this case, what is the interest of a treatment that increases capillary density without modifying cardiac function? It is however likely that an analysis of cardiac function beyond 10 days post-MI could give significantly different results.

As the number of LCZ treated animals were less than 6 per group, we made no statistical analysis of the data presented on Figure 9. However, we performed for this revised manuscript an additional experiment concerning the LCZ treated infarcted mice. We add to our previous data 4 mice treated with 60 mg/day/kg LCZ696, 2 mice treated with 6 mg/kg/day and 5 infarcted mice treated with saline. Now, the number of mice is sufficient to perform correct statistical analysis on the results. Figure 9 was modified and we added new results to the manuscript. Interestingly, LCZ606 treatment at both concentrations increased heart vascularization due to stimulation of endothelial cell proliferation but cardiac function was increased and heart remodelling decreased only in mice treated with the high dose of LCZ696. Interestingly, at high concentration LCZ696 stimulated also WT1^+^ cell proliferation in the ZI+BZ of infarcted hearts.

3) The authors should analyze whether or not LCZ696 directly stimulates the proliferation of resident mature endothelial cells and/or that of WT1+ cells.

We performed immunostainings to detect proliferating CD31^+^ cells and WT1^+^ cells on LCZ696-treated hearts. We represented the percentage of proliferation of endothelial or WT1^+^ cells on the Figure 9 (F and G) (i.e. we counted the BrdU^+^CD31^+^ and BrdU^+^Wt1^+^ cells and related the number by the total number of CD31^+^ or WT1^+^ cells, respectively). We added also representative pictures (Figure 9E), changed the figure legend and added these new results in the manuscript.

4) Results: the authors state that "first they determined whether ip BNP acted directly or indirectly on cardiac cells". But there is no single data set in this manuscript allowing to conclude that the observed effects are directly derived from endothelial actions of BNP. As they mention before, BNP acts on many types of cells and organs, and the observed effects could also be "indirect".

Yes it is true that BNP could act on endothelial cells indirectly, via other cardiac cells (cardiomyocytes or fibroblasts…). However in Figure 6D, we demonstrated that BNP acts directly on isolated adult endothelial cells and activates p38 MAP kinase on these cells. in vivo, we detected also p38 phosphorylation on endothelial cells in infarcted hearts after BNP treatment (please refer to Figure 6E and subsection “Stimulated proliferation of endothelial cells via p38 MAP kinase activation”).

5) Results paragraph three: plasma cGMP levels are a poor index of cardiac actions of BNP. It would be more meaningful to measure cardiac cGMP levels.

Results paragraph three : we measured cardiac cGMP levels and added this new information on Figure 1 (Figure 1B). The text was modified accordingly as well as the legend of the Figure 1. This result demonstrates that BNP acts directly on the heart.

6) Results paragraph four: it is strange to use the phosphorylation of phospholamban (PLB) as index of BNP activity. This manuscript focuses on angiogenesis. PLB is a regulatory protein in cardiomyocytes. Where is the link to endothelial regeneration?

Phospholamban (PLB) is a 24-27 kDa protein involved in the modulation of the reticulum Ca^2+^ ATPase (SERCA) ^3^. PLB inhibits Ca^2+^ uptake by SERCA. Phosphorylated form of PLB increases the affinity of SERCA for calcium and thus decreases the level of cytosolic calcium. It is well known than PLB is expressed by muscular cells such as cardiomyocytes and smooth muscle cells ^3, 4^. PLB is also expressed by non-contractile cells, such as endothelial cells (we precised this point in subsection “BNP direct action on cardiac non-myocyte cells after intraperitoneal injections”) ^5^. PLB modulates endothelium –dependent relaxation to acetylcholine^5^ and regulates via SERCA the integrity of the vascular barrier ^6^. Natriuretic peptides are able to increase cytoplasmic cGMP, which activates protein kinase G and leads to PLB phosphorylation. Thus, we used phosphorylation of PLB as a marker of BNP effect on the cardiac non-myocyte cells, including endothelial cells.

Previously, we showed that BNP via PKG activation and PLB phosphorylation, phosphorylates also p38 in Sca-1^+^ cells, which finally leads to their proliferation ^7^. Sca-1 protein is expressed on endothelial cells. However, the direct link between PLB phosphorylation and angiogenesis remains to be established.

7) Results paragraph four: BNP increased phosphorylation of PLB by nearly 200-fold in "non-myocyte cells" from the heart. Which cells are these? Is this fraction contaminated by cardiomyocytes? Which non-myocytes have such high PLB levels?

In non-myocyte cells, endothelial and smooth muscle cells express PLB as explained above (point 6). We isolated these NMCs from adult hearts using enzymatic digestion and retrograde perfusion directly in the hearts as described by Ackers-Johnson ^8^. We cannot exclude a small contamination with cardiomyocytes. However, we counted the cells and clearly discriminate between cardiomyocytes (large and medium cells) and NMCs (small cells). Thus, we estimated that the cardiomyocyte contamination is less than 5% among NMCs.

Furthermore, it is important to notice than all western blots depict on Figure 1 were performed using 25-30 microg proteins. This large amount of proteins allows to detect easily PLB and its phosphorylated form by western blot analysis. Finally, 200 fold increase is a relative unit, related to the level in NMCs isolated from saline treated hearts.

8) How were BNP plasma levels in BNP versus vehicle treated mice after MI? Did Entresto increase BNP plasma levels and to what extend?

It is difficult to measure BNP plasma level for several reasons:

– The half-life of BNP active form is 9-20 min.

– BNP processing is very complex as described in the Introduction. Different forms of BNP are present in the circulation (proBNP, BNP and NT-proBNP form) and the different kits available to measure BNP are not specific for the BNP active form. However, we could measure the NT-proBNP form in the plasma.

We didn’t measure the NT-proBNP form in our treated animals, because we sacrificed the mice 24h after the last BNP injection and we estimated that it is too late to highlight an increase of NT-proBNP form in BNP treated animals. However, we measured the cGMP in the heart and plasma (Figure 1B and C). Increased cGMP level is the consequence of increased BNP level.

In Entresto treated mice we didn’t measured the BNP level but this was done by others ^9-11^. We also measured the cGMP level in plasma of LCZ696 treated animals and detected a 3-fold increase of cGMP concentration 24h after LCZ696 treatment (60 mg/kg/day). This was explained in the Results.

9) Most in vitro and ex vivo studies were performed with NMCs. How many endothelial cells are contained in such heterogenous populations?

We isolated non-myocytes cells from neonatal and adult unmanipulated hearts to perform in vitro BNP-stimulations. As explained in the general comment above, isolated endothelial cells gradually die in culture. The presence of other non-myocyte cells allow to keep endothelial cells alive and to stimulate their proliferation.

At the end of the culture, the CD31^+^ cells represent 24±3% of the untreated and 27±4% of the BNP treated neonatal cells (n=32 different cell cultures). In adult NMC culture, only 12±1% of the cells are CD31^+^ cells 7 days after the onset of culture in untreated and BNP treated culture (n=16 different experiments).

In infarcted hearts, 3 days after MI, endothelial cells represent 21±3% of the NMCs in the ZI+BZ and 38±3% of the cells isolated from the RZ (n=16 different cell isolation). 10 days after MI, endothelial cells represent 25±2% of the NMCs of the ZI+BZ and 39±3% of the cells of the RZ (n=17 different cell isolation). The percentages didn’t vary between saline and BNP treated hearts but the number of cells is always increased in BNP treated animals, suggesting that BNP stimulated also the proliferation of other non-myocyte cells or protect these cells from apoptosis.

10) Some basic parameters are missing: how did BNP administration affect cardiac contractile functions as well as the infarct area and area at risk? Did exogenous BNP lower arterial blood pressure?

The analysis of cardiac functions of saline and BNP injected mice was already done and published in our first article (see Figure 7, Bielmann et al., 2014, Basic Research in Cardiology) ^2^. Cardiac contractility was 2 fold increased 4 weeks after MI in BNP treated animals (already published)^2^.

We also measured blood pressure in unmanipulated mice injected or not with BNP. We detected no difference (saline injected: 100.83 ±8.26, BNP 101 ±2.15 mm Hg). This suggests that most BNP effects after MI depends on cardiac rather than on vascular effects. We explain this now in more details in the first paragraph of the Results.

We measured blood pressure in LCZ treated mice and found no difference 10 days after surgery: saline MI: 103±18 mmHg; LCZ6 treated mice: 103±9 mmHg; LCZ60 treated mice: 109±9 mmHg. We added this result in the text.

11) Subsection “Stimulated proliferation of endothelial cells via p38 MAP kinase activation”: how does BNP, via NPR-A/cGMP-signaling, increase MAPK pp38? What is the signaling pathway and do the authors have any hint that this signaling pathway was also activated by BNP in vivo (in endothelial cells in situ)?

Previously, we showed that BNP via PKG activation and PLB phosphorylation, phosphorylates p38 in Sca-1^+^ cells, which finally leads to their proliferation ^7^. Phosphorylation of PLB was assessed as a marker of PKG activation. However, BNP activates also cAMP-dependent kinase (PKA), which is also able to phosphorylate PLB. PKG and PKA activation can both lead to p38 phosphorylation. We determined in the heart an increase of cGMP but we didn’t measure cAMP after BNP treatment. Thus, further experiments are needed on isolated CD31^+^ cells to precisely determine the signaling pathway involved.

We performed immunostainings against phosphorylated p38 on infarcted hearts treated or not with BNP. These are new results depict on Figure 6E which showed that endothelial and even WT1^+^ cells expressed pp38 in BNP-treated infarcted hearts.

12) Subsection "Increased vascularization in infarcted hearts after LCZ696 treatment". But in the corresponding Figure 9, there is no single data set showing “statistically significant effects of entresto”. The figure just shows some preliminary data and trends obtained with very few mice.

We performed a new experiment concerning the infarcted mice treated with Entresto. We add to our previous data 4 mice treated with 60 mg/day/kg LCZ696, 2 mice treated with 6 mg/kg/day and 5 infarcted mice treated with saline. Now, the number of mice is sufficient to perform correct statistical analysis on the results. Figure 9 was modified and we added new results to the manuscript.

13) Figure 1C: it is surprising that the basal levels of pPLB were so low (-). Normally, after MI in mice the endogenous ventricular expression levels of ANP and BNP significantly raise. Was there a difference in pPLB between sham and MI mice (vehicle treatments)?

As shown in Figure 1D, the pPLB/PLB ratio is lower in saline-treated infarcted hearts than in BNP-treated infarcted hearts.

Furthermore, we detected no increase of the pPLB/PLB ratio in infarcted hearts compared to sham (as shown in Author response image 1 in the RZ 3 days after MI). In contrast, we have a slight decrease which is consistent with the fact that patients with heart disease have in fact a deficit in functional active BNP and thus decreased level of PLB phosphorylation. As explained in the Introduction of the manuscript the balance between proBNP and active BNP is impaired in patients with heart diseases with higher levels of inactive proBNP and reduced levels of active BNP.

**Author response image 1. sa2fig1:** 

14) Figure 1D: which types of non-myocyte cells express such high pPLB levels and what is the functional meaning?

Phospholamban (PLB) is a 24-27 kDa protein involved in the modulation of the reticulum Ca^2+^ ATPase (SERCA) ^3^. PLB inhibits Ca^2+^ uptake by SERCA. Phosphorylated form of PLB increases the affinity of SERCA for calcium and thus decreases the level of cytosolic calcium. It is well known than PLB is expressed by muscular cells such as cardiomyocytes and smooth muscle cells ^3, 4^. PLB is also expressed by non-contractile cells, such as endothelial cells (we precise this point in the text)^5^. PLB modulates endothelium –dependent relaxation to acetylcholine^5^ and regulates via SERCA the integrity of the vascular barrier ^6^. Natriuretic peptides are able to increase cytoplasmic cGMP, which activates protein kinase G and leads to PLB phosphorylation. Thus, we used phosphorylation of PLB as a marker of BNP effect on the cardiac non-myocyte cells. In non-myocyte cells, endothelial and smooth muscle cells express PLB and BNP receptors. These cells can be directly stimulated by BNP, explaining thus the level of PLB phosphorylation in BNP-treated NMCs (Figure 1E now).

[Editors' note: further revisions were suggested prior to acceptance, as described below.]

The manuscript has been much improved but there are some remaining issues that need to be addressed before acceptance, as outlined below:The current title of the manuscript "Increasing heart vascularisation using brain natriuretic peptide stimulation of endothelial and WT1+ epicardial cells" has been changed from the version originally submitted ("Increasing heart vascularisation after myocardial infarction using brain natriuretic peptide stimulation of endothelial and WT1+ epicardium- derived cells.").It is important to clarify in the title (and the Abstract) that these findings apply to infarcted but not to healthy hearts, i.e. to post-infarct ischemic injury. The current title is no longer accurate as it does not mention the fact that the observations apply to cells in heart after myocardial infarction / injury. Thus, the title should be modified to "Increasing heart vascularisation after myocardial infarction using brain natriuretic peptide stimulation of endothelial and WT1+ epicardial cells ".

We agree to change the title of our manuscript in order to precise that the findings are related to infarcted hearts. This was also indicated in the Abstract.

References

Rignault-Clerc, S. *et al.* Functional late outgrowth endothelial progenitors isolated from peripheral blood of burned patients. *Burns* 39, 694-704 (2013).

Bielmann, C. *et al.* Brain natriuretic peptide is able to stimulate cardiac progenitor cell proliferation and differentiation in murine hearts after birth. *Basic Res Cardiol* 110, 455 (2015).

Colyer, J. Phosphorylation states of phospholamban. *Ann N Y Acad Sci* 853, 79-91 (1998).

Chen, W., Lah, M., Robinson, P.J. & Kemp, B.E. Phosphorylation of phospholamban in aortic smooth muscle cells and heart by calcium/calmodulin-dependent protein kinase II. *Cell Signal* 6, 617-630 (1994).

Sutliff, R.L., Hoying, J.B., Kadambi, V.J., Kranias, E.G. & Paul, R.J. Phospholamban is present in endothelial cells and modulates endothelium-dependent relaxation. Evidence from phospholamban gene-ablated mice. *Circ Res* 84, 360-364 (1999).

Balint, Z. *et al.* Double-stranded RNA attenuates the barrier function of human pulmonary artery endothelial cells. *PLoS One* 8, e63776 (2014).

Rignault-Clerc, S. *et al.* Natriuretic Peptide Receptor B modulates the proliferation of the cardiac cells expressing the Stem Cell Antigen-1. *Sci Rep* 7, 41936 (2017).

Ackers-Johnson, M. *et al.* A Simplified, Langendorff-Free Method for Concomitant Isolation of Viable Cardiac Myocytes and Nonmyocytes From the Adult Mouse Heart. *Circ Res* 119, 909-920 (2016).

Gu, J. *et al.* Pharmacokinetics and pharmacodynamics of LCZ696, a novel dual-acting angiotensin receptor-neprilysin inhibitor (ARNi). *J Clin Pharmacol* 50, 401-414 (2010).

Kompa, A.R. *et al.* Angiotensin receptor neprilysin inhibition provides superior cardioprotection compared to angiotensin converting enzyme inhibition after experimental myocardial infarction. *Int J Cardiol* 258, 192-198 (2018).

Menendez, J.T. The Mechanism of Action of LCZ696. *Card Fail Rev*2, 40-46 (2016).